# Environmental stability and phenotypic plasticity benefit the cold-water coral *Desmophyllum dianthus* in an acidified fjord

Kristina K. Beck [1,2✉], Gertraud M. Schmidt-Grieb[1], Jürgen Laudien [1], Günter Försterra[3,4], Verena Häussermann [3,5], Humberto E. González[6,7], Juan Pablo Espinoza[3,4], Claudio Richter [1,2,9] & Marlene Wall [1,8,9]

The stratified Chilean Comau Fjord sustains a dense population of the cold-water coral (CWC) *Desmophyllum dianthus* in aragonite supersaturated shallow and aragonite under-saturated deep water. This provides a rare opportunity to evaluate CWC fitness trade-offs in response to physico-chemical drivers and their variability. Here, we combined year-long reciprocal transplantation experiments along natural oceanographic gradients with an in situ assessment of CWC fitness. Following transplantation, corals acclimated fast to the novel environment with no discernible difference between native and novel (i.e. cross-transplanted) corals, demonstrating high phenotypic plasticity. Surprisingly, corals exposed to lowest aragonite saturation ($\Omega_{arag} < 1$) and temperature (T < 12.0 °C), but stable environmental conditions, at the deep station grew fastest and expressed the fittest phenotype. We found an inverse relationship between CWC fitness and environmental variability and propose to consider the high frequency fluctuations of abiotic and biotic factors to better predict the future of CWCs in a changing ocean.

[1] Alfred Wegener Institute Helmholtz Centre for Polar and Marine Research, Bremerhaven, Germany. [2] University of Bremen, Bremen, Germany. [3] Fundación San Ignacio del Huinay, Puerto Montt, Chile. [4] Pontificia Universidad Catolica de Valparaíso, Valparaíso, Chile. [5] Universidad San Sebastián, Puerto Montt, Chile. [6] Universidad Austral de Chile, Valdivia, Chile. [7] Centro FONDAP de Investigación en Dinámica de Ecosistemas Marinos de Altas Latitudes (IDEAL), Valdivia, Chile. [8] GEOMAR, Kiel, Germany. [9] These authors jointly supervised this work: Claudio Richter, Marlene Wall. ✉email: Kristina.Beck@awi.de

Scleractinian cold-water corals (CWCs) are important ecosystem engineers providing a three-dimensional habitat in cold and deep waters comparable to the complexity of shallow tropical coral reefs. CWCs sustain high levels of biodiversity and provide important nursery grounds for numerous benthic and fish species[1–3]. The distribution of CWCs is controlled i.a. by seawater carbonate chemistry, temperature, salinity, oxygen concentration, food availability and substrate topography[1,3–10]. Like tropical corals, CWCs cope with environmental variability through adaptive mechanisms, which makes them particularly vulnerable to rapid anthropogenic changes, especially ocean warming, acidification[11,12] and deoxygenation[10,13,14].

The physiological response of CWCs to changing temperature, pH and aragonite saturation ($\Omega_{arag}$) has so far been mainly investigated in laboratory studies under controlled conditions[15–20]. However, laboratory studies are usually conducted under constant conditions and do not consider the variability of environmental conditions that corals experience in their natural habitat. Some previous studies investigated the physiology of CWCs in situ[21–23], but only few studies considered the seasonal differences in biotic and abiotic parameters or the small-scale environmental heterogeneities in the habitat of CWCs[24,25]. The few available in situ measurements show that temperature, salinity, oxygen and pH vary seasonally and even daily in response to tides, internal waves and advection[9,26–29], suggesting that the environment of CWCs is far less uniform than previously assumed. However, the lack of in situ data limits our ability to assess the extent and ability of CWCs to cope with fluctuating environmental conditions. The physiological response of marine organisms to variable conditions may differ from their response to constant environmental conditions, as has been shown for phytoplankton and mussels[30–32]. This suggests that the physiological response of CWCs[21,24] and other organisms[31,32] in laboratory experiments acclimated to stable environmental conditions may not match their physiological performance in natural, varying environments. Organisms from an environment with greater variability may have enhanced tolerance to changing environmental conditions[33]. Therefore, we need to improve our understanding of the impact of natural environmental variability on the resilience of corals. Several tropical coral species have already been shown to be more resistant to heat stress and less susceptible to bleaching in thermally variable environments[33–36]. On the other hand, corals from more stable environments may be less resistant to climate change if the corals have adapted to their natural habitat[37]. Thus, it is similarly important to know whether populations are adapted to local conditions and if this also determines their response to future changes.

In order to assess the resilience of CWCs to future climate changes, it is important to first understand their physiological performance under present-day in situ conditions and environmental variability. Therefore, long-term studies of oceanographic conditions in situ are paramount to assessing natural environmental variability. Species living under different environmental conditions enable us to study their acclimatisation and adaptation potential and ability to deal with natural variability. Acclimatisation is the ability of an organism to change its physiological performance in response to changes in environmental conditions without genetic changes (phenotypic plasticity). By contrast, adaptation represents changes in the genome in response to environmental conditions over multiple generations through natural selection[38]. A good opportunity to study the response and acclimatisation potential of corals to changing environmental conditions in situ are reciprocal transplantation experiments[39,40]. A transplantation experiment can show either (1) adaptation if the organism's performance depends on its origin and is not influenced by the environment, (2) local adaptation if the

organism's performance depends on both origin and environmental conditions, or (3) acclimatisation if the organism's performance depends on the environment, regardless of its origin[39–41]. Reciprocal transplantation experiments often show local adaptation of organisms, but sometimes native organisms at the site of origin are not better adapted than organisms that were transplanted from different environments[39,40]. Several studies transplanted tropical corals from their natural habitats to stations with contrasting abiotic conditions[37,41,42], but only two reciprocal transplantation experiments with CWCs have been conducted so far[23,43].

The aim of this study was to investigate the physiological performance and acclimatisation potential of CWCs to changing in situ environmental conditions. Therefore, we conducted a year-long reciprocal transplantation experiment with the CWC *Desmophyllum dianthus* in Comau Fjord (northern Chilean Patagonia, Fig. 1a) in order to distinguish between long-term adaptation and short-term acclimatisation to the local environment. We took advantage of the occurrence of *D. dianthus* in contrasting environments in the stratified Comau Fjord, with low salinity, high pH and oxygen concentrations in surface waters and marine conditions, but low pH and oxygen levels in deeper waters[44–48]. Corals were collected at opposing ends of horizontal and vertical environmental gradients[49] (head vs. mouth and shallow vs. deep) and reciprocally transplanted in order to study their physiological responses to contrasting environmental conditions (Fig. 1b). The vertical gradient persists with the strong environmental differences described[44,47,49], whereas the horizontal gradient is strongest in the productive summer season, but influenced by mixing in autumn and winter (Supplementary Data 1 and Supplementary Fig. 1). Over a one-year period, coral fitness traits were evaluated every 3 to 4 months focusing on calcification, respiration and tissue composition. In addition, tissue coverage was measured at the beginning and end of the experiment. We aimed to better understand the drivers of acclimatisation and adaptation to local environmental parameters by (1) characterising differences in environmental parameters in the natural habitat of *D. dianthus*, (2) measuring coral physiological parameters along environmental gradients and between seasons, and (3) correlating environmental conditions with *D. dianthus* performance. Here we show that *D. dianthus* benefits from stable environmental conditions in the deep waters of the fjord and is able to acclimatise quickly to a new environment after transplantation.

## Results

**Environmental variability**. The water temperature at 20 m water depth in Comau Fjord shows a mean annual temperature of $12.5 \pm 0.9\,°C$ with high-frequency fluctuations at all stations (Fig. 2a and Supplementary Data 1). The highest variability was found in austral summer and autumn with daily temperature fluctuations of up to $3.7\,°C$ and a maximum temperature of $16.6\,°C$ (Fig. 2a and Supplementary Data 1). CTD data from station X showed that temperature and salinity co-varied strongly throughout the year (Supplementary Fig. 2). Temperature and salinity fluctuations were correlated, but the direction and strength differed with the season, changing from a positive relationship from spring until autumn to a negative one in winter (Supplementary Fig. 2). Salinity regularly fluctuated between 31.5 and 32.5 but occasionally also reached below 30 in winter. In contrast, water temperatures at 300 m water depth were lower and showed much less variability throughout the year with a mean temperature of $11.4 \pm 0.2\,°C$ (Fig. 2a and Supplementary Data 1). In the deep waters of Comau Fjord, water temperatures increased only slightly and fluctuated more in austral winter.

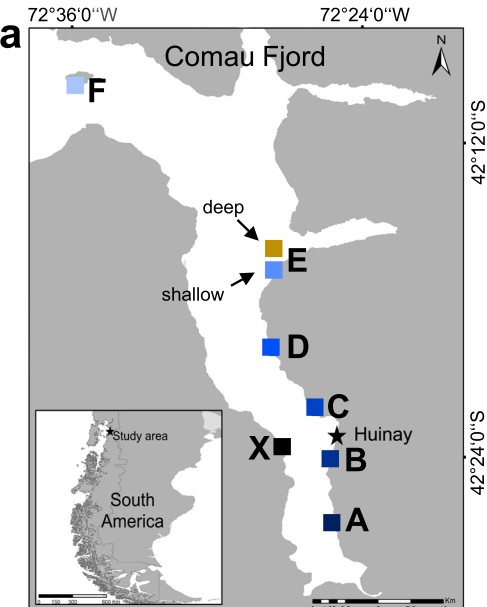

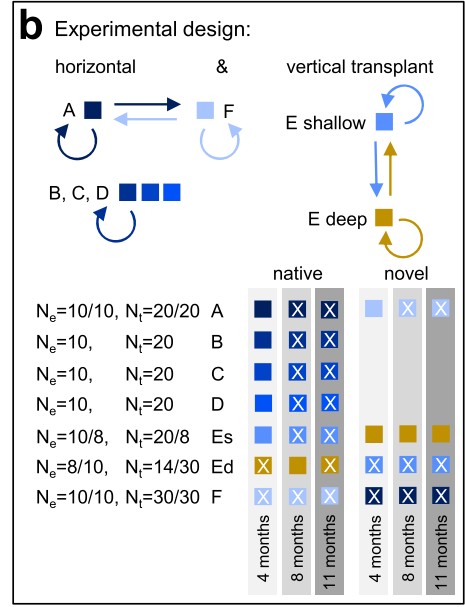

**Fig. 1 Experimental design and coral sampling scheme. a** Coral sampling stations in Comau Fjord, Chile: six stations at 20 m water depth (A–F shallow, blue) and one station at 300 m water depth (E deep, yellow). The research station in Huinay is located between stations B and C (star). The CTD was deployed at 25 m water depth at the station X. **b** The experimental design includes vertical and horizontal reciprocal transplantation of novel (i.e. cross-transplanted) corals between the shallow stations A and F as well as between shallow (E shallow: Es) and deep (E deep: Ed), where colours indicate the station of origin. Corals collected at stations B, C and D were only returned to their respective native station. One subset of corals (experimental corals) was used repeatedly for calcification and respiration measurements over the entire experimental period (i.e. after 4, 8 and 11 months; number of individuals $N_e$: 8–10 biologically independent samples per station and sampling time point). A second subset of corals (tissue corals) was sampled for biomass analysis after, 8 and 11 months ($N_t$: 6–10 biologically independent samples per station and sampling time point; see Supplementary Methods and Results). The tissue corals were initially sampled for the same time points as experimental corals, but due to logistical problems could only be obtained for the time points and stations marked with an X.

However, in austral winter, water temperatures were higher at 300 m compared to 20 m water depth due to surface cooling and convective mixing of the upper water column. The salinity was higher at the deep station compared to all shallow stations, whereas the oxygen concentration, $pH_T$ and $\Omega_{arag}$ were lower at 300 m compared to 20 m water depth but followed the same seasonal pattern (Fig. 2b–e, Supplementary Fig. 3 and Supplementary Data 1).

**Calcification and respiration rates of native and novel corals**. Calcification and respiration rates of both native and novel (cross-transplanted) corals of *D. dianthus* were significantly higher at 300 m compared to 20 m depth (LMM; Es-Ed: *p* value <0.001, Fig. 3a, b, Supplementary Table 1 and Supplementary Data 2, 3). In shallow waters along the fjord, calcification rates were higher at the mouth of the fjord compared to the head (LMM; A–F: *p* value <0.001; Fig. 3a, Supplementary Table 1 and Supplementary Data 2, 3) and differed between seasons with lower calcification rates in austral winter (August) compared to austral summer (January) and autumn (May; LMM; Jan-Aug and May-Aug: *p* value <0.001, Fig. 3a, Supplementary Table 1 and Supplementary Data 2, 3). Respiration rates at shallow stations were also higher at the mouth of the fjord (LMM; A–F: *p* value = 0.011; Fig. 3a, Supplementary Table 1 and Supplementary Data 2, 3), but highest at station C (LMM; e.g. A–C: *p* value <0.001; Fig. 3a, Supplementary Table 1 and Supplementary Data 2, 3), and also differed between seasons with higher respiration rates in austral summer compared to autumn and winter (LMM; e.g. Jan-May: *p* value = 0.007 and Jan-Aug: *p* value = 0.038; Fig. 3a, Supplementary Table 1 and Supplementary Data 2, 3).

Novel corals at the shallow stations and the deep station adjusted their calcification rates to the novel environmental conditions of the respective station (LMM; transplant: *p* value = 0.257, Fig. 3a, Supplementary Table 1 and Supplementary Data 2, 3). However, both native and novel corals at the deep station changed their morphology throughout the experiment and had trumpet-shaped calyxes at the end of the experiment (Supplementary Fig. 4b). Novel deep corals transplanted from shallow waters were also able to expand their tissue surface area and biomass to a similar extent as the native deep corals (Supplementary Figs. 4–6). Transplantation of corals between stations had also no effect on respiration rates as novel corals at all four stations showed the same respiration rates as native corals at the same station (LMM; transplant: *p* value = 0.562 Fig. 3b, Supplementary Table 1 and Supplementary Data 2, 3). Therefore, novel corals acclimatised fast to novel environmental conditions after transplantation in shallow and deep waters and showed the same physiological response as the native corals at the respective stations.

The calcification rates of *D. dianthus* correlated with the mean annual and seasonal $pH_T$ and $\Omega_{arag}$ in Comau Fjord at the different stations (Supplementary Fig. 7a–d). However, calcification rates were highest in undersaturated conditions. Therefore, we performed a model selection with all environmental parameters including $pH_T$, $\Omega_{arag}$, mean seasonal temperature, mean seasonal temperature variability, salinity and oxygen concentration. This multifactorial analysis showed that mean seasonal temperature and mean seasonal temperature variability were the two most important factors explaining 55 % of the calcification data (adjusted $R^2 = 0.555$, Fig. 4). Temperature variability was negatively related to calcification rates of *D. dianthus* in Comau Fjord as calcification rates were highest at the deep station with the lowest temperature variability.

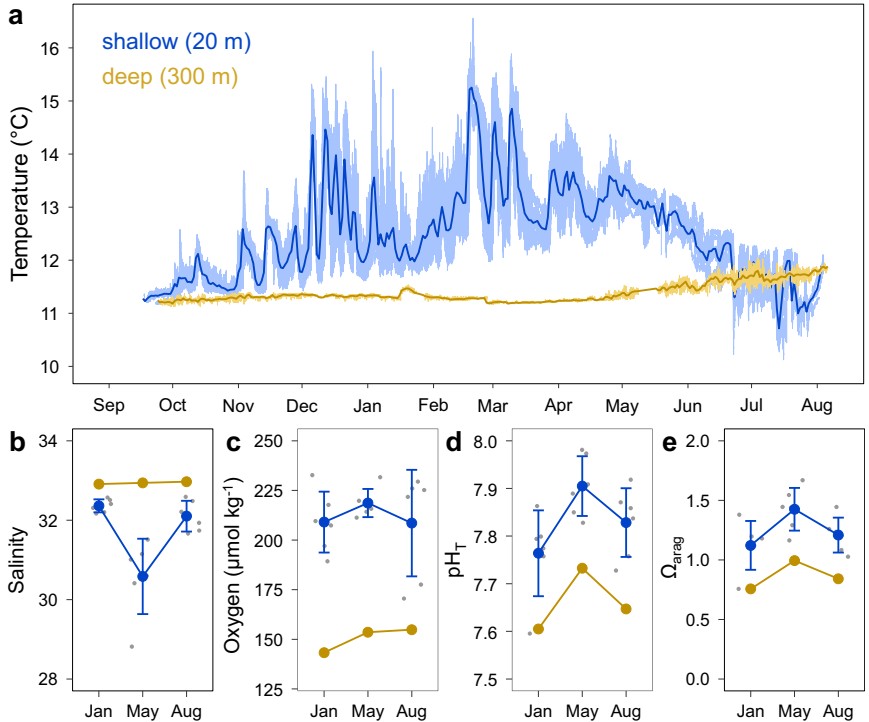

**Fig. 2 Seasonal environmental conditions in shallow and deep waters of Comau Fjord, Chile. a** Mean water temperature at all shallow stations at 20 m water depth (blue) and at the deep station at 300 m water depth (yellow) between September 2016 and August 2017. The light blue shaded area is the raw temperature of all six shallow stations and the yellow shaded area is the raw temperature of the deep station. The blue and yellow lines are the daily mean temperature data for all shallow and deep stations, respectively. **b** Salinity and **c** oxygen from CTD, **d** seawater $pH_T$ and **e** aragonite saturation ($\Omega_{arag}$) were calculated from TA and DIC in austral summer (January), autumn (May) and winter (August). **b**–**e** Mean ± standard deviation for all shallow stations at 20 m water depth (A–F, $N = 6$ independent samples) are shown in blue and conditions at the deep station at 300 m water depth (Ed, $N = 1$) in yellow (data points in grey).

## Discussion

This horizontal and vertical reciprocal transplantation experiment with CWCs assessed the seasonal physiological performance of the CWC *Desmophyllum dianthus* between two spatially close but strongly contrasting environments. Corals were transplanted between an aragonite saturated, thermally variable shallow environment and an aragonite undersaturated, thermally stable deep environment in Comau Fjord, Chile. Unexpectedly, we found the fittest (i.e. fastest-growing) *D. dianthus* in deep waters characterised by aragonite undersaturation, but also stable environmental conditions. Following transplantation, *D. dianthus* showed remarkably fast acclimatisation to the novel environment with no discernible differences between native and novel corals despite different environmental conditions at their original location. This underscores the ability of *D. dianthus* to accommodate a broad range of habitats and may help explain the great success of this cosmopolitan CWC species.

The low pH and aragonite undersaturation did not reduce the physiological performance and fitness (calcification) of *D. dianthus* (Supplementary Fig. 7a–d), explaining the so far enigmatic occurrence of dense coral banks in deep waters of Comau Fjord[44]. CWC reefs in other areas (e.g. Gulf of Mexico, SW Australia) dominated by *L. pertusa* have also been found close to or below the aragonite saturation horizon[6,7,9]. However, *D. dianthus* occurs at lower $\Omega_{arag}$, pH, TA and DIC values than previously reported for CWC habitats in the NE Atlantic Ocean and Mediterranean Sea[8,50]. In austral summer (January), the pH and aragonite saturation in shallow waters of the fjord head were as low as in deep waters at station Ed, presumably due to increased run-off and decomposition of terrestrial organic matter, showing

that the observed differences in calcification rates are not related to differences in aragonite saturation (Supplementary Fig. 7a–d). As we found the fittest specimens at the highest DIC values in Comau Fjord, this does not support the hypothesis that the occurrence of healthy CWC reefs is prevented by high DIC[8]. While breakage and dissolution of bare coral skeletons occur under aragonite undersaturation[15,51], there is growing evidence that carbonate chemistry is not as important for live CWCs as has long been thought. Several laboratory and field studies have confirmed the capability of CWCs to calcify and survive at aragonite undersaturated conditions[15,24]. Even though it was not statistically significant, a similar trend of highest calcification rates under aragonite undersaturation was also found in a long-term laboratory experiment with *L. pertusa*[52], which is explained by the overall good physiological conditions of the corals due to regular feeding. However, the fact that CWCs are able to maintain calcification despite low pH in situ and in laboratory experiments does not explain why the fittest individuals of *D. dianthus* are found at 300 m depth in Comau Fjord, showing an almost twice as high calcification rate.

As elevated temperatures coincide with elevated metabolic rates[17,53,54], we expected to find higher CWC calcification rates at shallow stations, where the mean seasonal temperature was up to 1.9 °C higher than at 300 m depth. However, environmental variability can modulate performance[55,56] and may lead to reduced performance in shallow waters as the highest temperature fluctuations at the head of the fjord coincide with the lowest coral performance along the horizontal gradient (Figs. 3 and 4). The macrotidal environment of Comau Fjord features tidal ranges of up to 7.5 m[46] and associated temperature and salinity

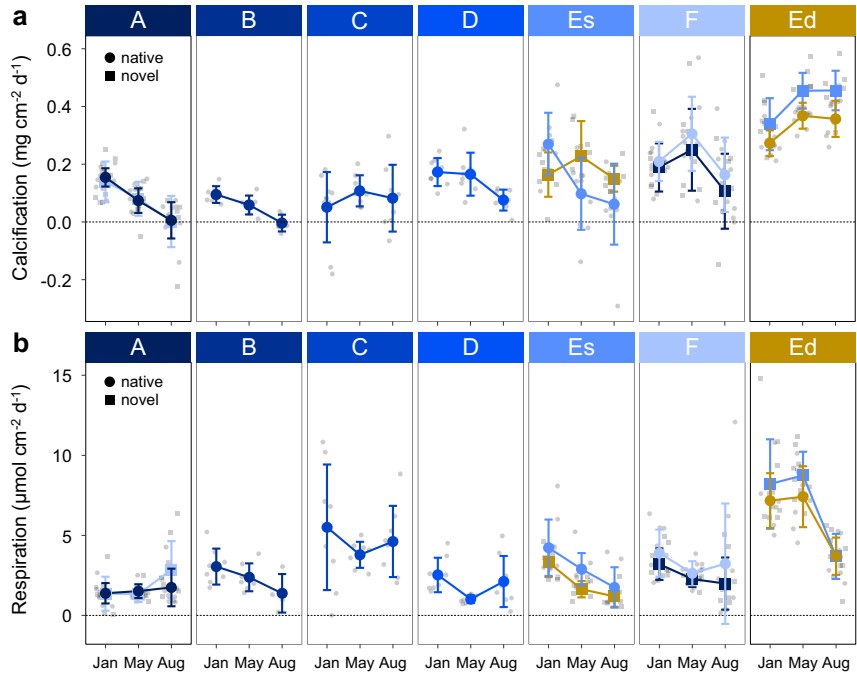

**Fig. 3 Seasonal physiological parameters of native and novel *Desmophyllum dianthus* in Comau Fjord, Chile. a** Calcification and **b** respiration rates of *D. dianthus* (mean ± standard deviation) at six stations at 20 m water depth along the fjord from head to mouth (A–F) are shown in blue and at one station at 300 m water depth (Ed) in yellow (data points in grey). Native corals (circles) were re-installed at the same station after collection in September 2016 and novel corals (squares) were cross-transplanted between the shallow stations at the head (A) and the mouth of the fjord (F) and between shallow (Es) and deep (Ed). Calcification and respiration rates were measured after 4, 8 and 11 months (January, May and August 2017) using the same individuals in each season ($N = 4$–10 independent samples) and standardised to the tissue-covered surface area of the corals at the beginning of the respective growth period. Respiration rates were measured at a standardised temperature of 12.75 ± 1.44 °C, representing water temperatures of shallow stations at the end of each season (January: 14.16 ± 0.43 °C, May: 12.19 ± 0.63 °C, August: 11.76 ± 1.63 °C) and not at in situ temperatures of each station.

fluctuations in shallow waters with daily swings of more than 3 °C and short temperature peaks beyond 16 °C in summer (Fig. 2a and Supplementary Fig. 2). Similar short-term temperature fluctuations were also measured by Rossbach et al.[23] in Comau Fjord, in neighbouring Reloncaví Fjord[57] and in other deep CWC habitats[26,28,58,59]. Thus, even in CWC habitats that have long been regarded as stable environments, short-term variabilities are ubiquitous and of ecological importance in macrotidal environments. They affect not only temperature and salinity[4,9,26,28,60], but also oxygen concentration[4,61–63], pH_T and nutrients[27]. As all conservative oceanographic parameters co-vary with temperature and potentially affect the performance of corals, we used temperature as a proxy for environmental variability in this study.

So far, no studies addressed the ability of CWCs to cope with short-term environmental fluctuations. Data are accumulating on the range of environmental conditions in the natural habitats of CWCs that exhibit high-frequency fluctuations[6,7,9,27,29], underscoring the dynamics that CWCs can be exposed to in their natural habitat, as well as the temporal and spatial range relevant to them. However, we often miss corresponding data on the physiological performance of corals. Only few studies investigated the short-term effect of elevated temperatures on CWCs[54,59]. Temperatures of up to 15 °C, that may be comparable to daily temperature fluctuations in their natural habitat, lead to increased metabolic activity[54]. In long-term studies under constant conditions, the effect of elevated temperatures on the performance of different CWC species varies and differs between locations[54] as well as species[53,64], ranging from a positive effect on calcification to no effect or even a negative response (Table 1). Similar variable results are found for the effect of pH with differences between locations[16], species[65] as well as short- and long-term

exposure[19,52]. Reduced pH either reduces calcification rates of CWCs, has no effect on their calcification or even slightly enhances long-term calcification rates of CWCs[52] (Table 1). CWCs in regions with large oxygen variability[61,62] are able to tolerate low oxygen concentrations[10,14], but only for a short period of time under experimental conditions[63,66]. However, studies investigating the effect of environmental factors other than temperature and pH as well as experiments with multi-factorial design with CWC are still scarce (except for[15,17,18]). Similarly, most studies used constant conditions and neglected naturally occurring strong and short-term fluctuations relevant to individual organisms[67,68]. However, this may hamper our understanding of the performance of organisms, contribute to observed response heterogeneity and limit our predictions for the future of the ocean[69,70]. As extreme events are expected to increase in the future[71,72], higher environmental variability may expose CWCs to more stressful conditions.

There is a growing body of literature addressing environmental sensing techniques with high temporal resolution and high-lighting the ubiquity of environmental variability at numerous temporal and spatial scales as well as at remote locations[73,74]. All emphasise their physiological relevance and initial experiments demonstrate their ecological significance and ability to influence organism responses[75–77]. Several studies on tropical corals revealed that environmental variability can improve their performance and render them more stress tolerant[35,36,78,79] (but see also[55,56]). However, our study clearly indicates that this may not be the case for CWCs as environmental variability negatively correlates with coral performance (Fig. 4). It indicates that high environmental variability entails energetic costs in *D. dianthus*, compromising calcification in shallow waters, whereas deep

corals may need less energy to compensate for stress. However, it still needs to be elucidated in more detail whether environmental fluctuations have a direct or only indirect effect on coral fitness, e.g. by influencing their food availability.

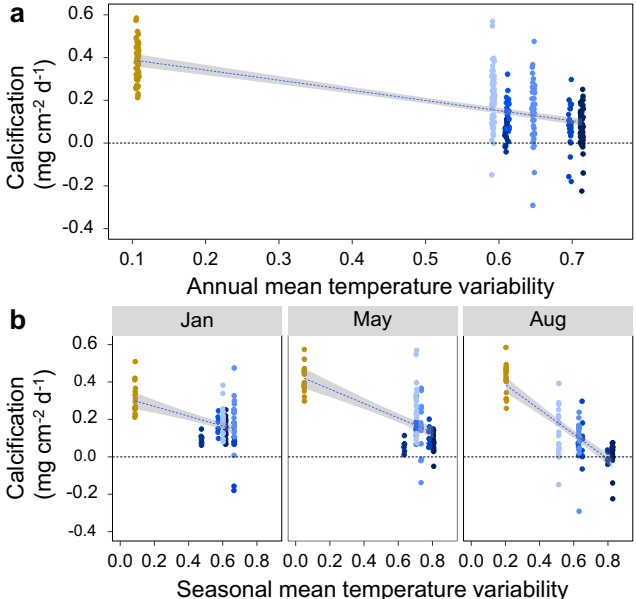

**Fig. 4 Relationship of calcification rates of *Desmophyllum dianthus* and mean temperature variability at sampling stations in Comau Fjord, Chile. a** Combined data of all three seasons (*N* = 19–57 independent samples) and **b** seasonal comparison of calcification rates with mean temperature variability (*N* = 6–20 independent samples). Calcification rates of *D. dianthus* at six stations at 20 m water depth along the fjord (A–F) are shown in blue and at one station at 300 m water depth (Ed) in yellow. Calcification rates of native and novel corals were measured after 4, 8 and 11 months (January, May and August 2017) using the same individuals in each season. Note that native and novel corals at each station are combined in this graph. Adjusted *R*² = 0.555.

Due to high environmental variability in shallow waters of Comau Fjord, corals are likely to feed less during periods of elevated temperatures, as suggested by the low prey capture rates of *Lophelia pertusa* and *Madrepora oculata* in laboratory experiments at higher temperatures (17 °C)[80], which are probably a consequence of lower polyp activities. Fluctuating physico-chemical conditions may also alter zooplankton communities both in abundance and composition[81,82]. For instance, changes in salinity may result in the reduction in coastal zooplankton as osmotic stress can be lethal for some zooplankton groups[82]. In deep waters (300 m) of Comau Fjord, zooplankton abundance and biomass are low throughout the year and show only low diel variations[48]. At shallower depths (0–200 m), abundance and biomass are often an order of magnitude higher and strongly affected by diel vertical migration[48], supporting a higher food availability for shallow corals. Unlike calcification rates, which integrate over months, respiration rates were determined over hours and are thus not entirely comparable. However, they provide insights into the current metabolic potential of the corals and indicate a generally elevated metabolic activity in deep corals (supported by higher calcification rates and biomass; Fig. 3a, Supplementary Results, and Supplementary Fig. 6), possibly fuelled by higher food intake[83,84] (but also see Supplementary Discussion for temperature effects on respiration rates of deep corals). While the higher food availability appears to contradict the zooplankton data, it has to be taken into account that other factors (e.g. swarming behaviour, zooplankton aggregations near walls, micronekton, etc.) may play an important role in the food supply to CWCs. Similarly, factors such as competition for food from a diverse benthic community, particularly found in shallow waters, may further deplete the available abundant zooplankton community. It is also plausible that recurrent disturbances caused by high-frequency environmental fluctuations as well as mobile benthic organisms influence the behaviour of the corals, e.g. by inducing the retraction of their tentacles and thus a polyp inactivity[80], which limits their feeding time. Another important aspect is the energy density as well as energy gain from different zooplankton groups that may also contribute to differences in performance.

**Table 1 Effect of *elevated temperature* and *reduced pH/elevated pCO2* on calcification rates of cold-water corals.**

| Effect | Duration | Region | Species and references |
|---|---|---|---|
| **Elevated temperature** | | | |
| positive | short-term | Mediterranean Sea | *Dendrophyllia cornigera*[112], *Dendrophyllia ramea*[112] |
| | long-term | Mediterranean Sea | *Lophelia pertusa*[53], *Madrepora oculata*[53], *Dendrophyllia cornigera*[64,113] |
| | | NE Atlantic Ocean | *Lophelia pertusa*[18] |
| none | long-term | Mediterranean Sea | *Lophelia pertusa*[80], *M. oculata*[80], *Desmophyllum dianthus*[64] |
| | | NE Atlantic Ocean | *Lophelia pertusa*[15] |
| negative | long-term | Mediterranean Sea | *Desmophyllum dianthus*[17], *Dendrophyllia cornigera*[112], *Dendrophyllia ramea*[112] |
| **Reduced pH/elevated pCO₂** | | | |
| none | short-term | Mediterranean Sea | *Lophelia pertusa*[114], *Madrepora oculata*[114,115], *Desmophyllum dianthus*[24] |
| | | NE Atlantic Ocean | *Lophelia pertusa*[16,116] |
| | | Gulf of Mexico | *Lophelia pertusa*[19] |
| | long-term | Mediterranean Sea | *Lophelia pertusa*[114,117], *Madrepora oculata*[114,117], *Desmophyllum dianthus*[17,24], *Dendrophyllia cornigera*[24,65], *Caryophyllia smithii*[24] |
| | | NE Atlantic Ocean | *Lophelia pertusa*[15,52], *Desmophyllum dianthus*[118] |
| | | SW Pacific Ocean | *Solenosmilia variabilis*[119] |
| negative | short-term | Mediterranean Sea | *Madrepora oculata*[120] |
| | | NE Atlantic Ocean | *Lophelia pertusa*[52,121] |
| | | NE Pacific Ocean | *Lophelia pertusa*[122] |
| | | Gulf of Mexico | *Lophelia pertusa*[16,63] |
| | long-term | Mediterranean Sea | *Desmophyllum dianthus*[65] |
| | | NE Atlantic Ocean | *Lophelia pertusa*[18] |
| | | SE Pacific Ocean | *Desmophyllum dianthus*[20] |
| | | Gulf of Mexico | *Lophelia pertusa*[19] |

Duration of exposure: short-term <3 months and long-term ≥3 months.

A laboratory study by Maier et al.[84] clearly indicates that prey size is important. With a diet of small fjord zooplankton, *D. dianthus* requires a minimum of 700 individuals per day to balance metabolic needs, while the addition of one euphausiid provides a positive scope for growth[84]. Euphausiids are known to form dense swarms in Chilean fjords[85,86], but data from the immediate vicinity of the corals are missing. Euphausiids are underrepresented in the study by Garcia-Herrera et al.[48] as they are active swimmers capable of escaping plankton nets at low tow speeds. Mysids account for up to 70% of the zooplankton volume and abundance in deep waters of Comau Fjord in all seasons and may provide a reliable food source for deep corals throughout the year[48] with potentially similar energy gain as derived from euphausiids[84,87]. Thus, a difference in zooplankton composition may compensate for a large difference in zooplankton abundance between depths, and together with other drivers (high-frequency fluctuations, competition, behavioural adaptation), contribute to differences in energy gain.

The present study shows remarkable phenotypic plasticity of *D. dianthus* after transplantation, underscoring its potential to acclimatise to local environmental conditions in Comau Fjord (Fig. 3). Seasonal calcification data show that novel corals acclimatised quickly in less than four months without any subsequent change throughout the remainder of the study. Population genetic data support the lack of local adaptation with gene flow along both the horizontal and vertical axes of the fjord[49]. Similar results of local acclimatisation have been shown for calcification rates of the gorgonian coral *Antillogorgia bipinnata*[88] and several tropical coral species[89,90] after reciprocal transplantation. The morphological change of the native and novel deep corals on the pulley in the water column is also a sign of acclimatisation to the environmental conditions in deep waters. However, we cannot say to what extent biological factors (e.g. supposedly lower predation) or environmental factors (e.g. stronger currents) may also have played a role compared to the corals on the fjord wall. For example, it was observed that *L. pertusa* colonies grow in different shapes depending on the hydrodynamic conditions[91]. However, some reciprocal transplantation experiments clearly revealed local adaptation[37,92] or a combination of local adaptation and phenotypic plasticity depending on the investigated traits[35,41,78]. The main traits measured in the present study (calcification and respiration) show clear acclimatisation of *D. dianthus* to novel environmental conditions but other traits may also provide insights into potential local adaptation. The low tissue coverage of many shallow corals (mainly at stations B, C and Es) indicates that these corals have a reduced scope for somatic growth (Supplementary Figs. 4–6). In addition, the low tissue cover is associated with a higher infestation with endolithic photoautotrophic organisms[93–95] (Supplementary Fig. 4), which negatively affects their septal linear extension rates[95]. This may be an additional stressor for shallow corals and contribute to a potentially reduced fitness[95] as the defence against infesting organisms and maintenance of the skeletal integrity requires increased energy expenditure. However, this likely did not affect deep corals to the same extent, which were completely covered with tissue and therefore protected throughout the duration of the experiment. Deep corals transplanted to shallow were able to maintain their tissue-covered surface area without tissue retraction (Supplementary Figs. 4 and 5). Whether this is a real local adaptation or a delayed response, potentially caused by the availability of enhanced energy reserves, needs to be elucidated.

The large tissue-covered surface area of native deep corals and the tissue surface area expansion of novel deep corals also protected the coral skeletons from dissolution at aragonite undersaturation. As the buoyant weighing method measures net calcification (growth minus dissolution), the higher calcification rates in deep waters may therefore be partly due to lower dissolution rates. Novel corals at the deep station invested considerably more energy into somatic growth compared to native deep corals. Altogether, deep corals not only have 1.7-fold higher respiration and 2.3-fold higher calcification rates than shallow corals, but they also have a higher scope for somatic growth and invest 6.7-times more into the build-up of tissue biomass (Supplementary Results and Supplementary Fig. 6) and presumably also into energy reserves. This means that firstly, there may be more energy available at the deep station and secondly, energy may be channelled differently as additional energy is used to build up biomass rather than for calcification. In contrast, somatic growth is clearly sacrificed at the shallow stations, maybe to maintain reproductive output[96]. As reproduction requires substantial energy[25], low tissue coverage and calcification rates of shallow corals could potentially indicate an energetic trade-off. In addition, decreasing calcification rates may not solely be linked to decreased seasonal temperatures, but indicate that more energy is channelled into reproduction than into other traits as *D. dianthus* is actively reproducing in shallow waters of this fjord in austral winter[97]. The results of the present study indicate that deep corals generally have more energy available and are potentially also more fecund as proposed by Feehan et al.[97], but nothing is known about their reproductive cycle so far. Therefore, future studies on coral energetics should also include the reproductive cycle, in order to better understand the interplay of traits and potential fitness consequences for the corals as well as the whole population.

While somatic growth provides hints towards local adaptation, it warrants further investigations of other traits when studying the acclimatisation potential of CWCs to a new environment. Some previous studies on tropical corals revealed these possible different trait responses[41,78]. For instance, the biochemical signature together with gene expression[41] could provide more detailed insights. Additional studies are also necessary to identify if somatic growth of deep corals in shallow will be reduced after a longer period of time and whether calcification and respiration rates continue to show local acclimatisation. Temporal effects are known from tropical studies, where some traits may acclimatise faster than others. For instance, *Porites astreoides* showed higher fitness (calcification and energy reserves) in the natural environment (local adaptation) but only after twelve months of the experiment and not yet after 6 months[37].

The physiological performance of *D. dianthus* in Comau Fjord demonstrates the strong phenotypic plasticity of this cosmopolitan CWC. While somatic growth indicates local adaptation, calcification and respiration are clearly driven by the environment. Unexpectedly, the fittest phenotype was found in deep waters of the fjord with aragonite undersaturated conditions. Although depth-related differences in food availability and bioerosion could not be accounted for, our results indicate that environmental variability plays a stronger role than aragonite saturation in governing CWC calcification in this macrotidal system, where shallow corals are exposed to stronger environmental fluctuations. High fluctuations in other regions may thus be one reason that limits the ability of CWCs to emerge from the deep and conquer shallow environments. However, this warrants further investigations into the relative role of individual parameters and their ability to limit the performance of CWCs. It is important to consider environmental variability in CWC habitats in future physiological studies when investigating their potential vulnerability to changing environments, rather than assuming constant conditions. While we now have a better idea of the environmental conditions in the natural habitats of CWCs, important information on the physiological performance of corals is still missing. Such data are crucial to better understand the

well-being of corals and their response to future changes, as environmental variability (especially temperature, salinity and pH) is likely to increase in the future, exposing CWCs to more stressful conditions.

## Methods

**Study site and organisms**. Comau Fjord in the northern part of Chilean Patagonia has a total length of about 45 km, a width of 2–8.5 km and a maximum depth of almost 500 m near the mouth[98]. It is characterised by a high tidal range of up to 7.5 m[46]. Throughout the Chilean fjord region, high precipitation and glacial melt lead to substantial freshwater runoff, causing strong stratification with a superficial low salinity surface water layer down to 7–15 m depth and a marine subsurface layer below[44,46,48]. As a result of high inputs of terrigenous organic material and its subsequent degradation at depth, the marine layer is low in oxygen (as low as 40–50% saturation) and pH (<7.7)[45,48].

In Comau Fjord, the cosmopolitan, azooxanthellate CWC species *Desmophyllum dianthus*, typically a deep-sea species, is found at exceptionally shallow depths of up to 15 m[46,93]. It is the most abundant coral species in the Chilean fjord region, with densities of more than 1500 specimens m$^{-2}$ below 25 m depth and provides habitat for a diverse benthic community[3,93,99]. *Desmophyllum dianthus* is a pseudo-colonial species with aclonal individuals that can grow on top of each other[93], but tend to be solitary in shallow waters of the fjord. In Comau Fjord, *D. dianthus* mainly grows under overhangs, the underside of rocks and on the steep fjord walls with the calyx oriented downward[93,100,101], presumably to prevent the negative effects of high sedimentation rates in this region. The distribution of *D. dianthus* across a wide range of environmental conditions provides a rare opportunity to study the response of CWCs to contrasting conditions.

**Environmental data**. At each coral station (Fig. 1a), the water temperature was recorded (Tidbit v2 logger, ONSET computers, Bourne, USA; 0.2 °C resolution; attached to one of the coral plates) in 15 min intervals over the whole study period (September 2016–August 2017). Salinity was measured with a CTD (SBE 19 plus, V2 SeaCAT profiler, Sea-Bird Scientific, Bellevue, USA; internal sensors: temperature, conductivity, pressure; external sensor: oxygen sensor SBE 43, Sea-Bird Scientific, Bellevue, USA) once during each season (January, May and August 2017). In addition to the coral stations, temperature and salinity were also measured every 30 min with a CTD (AML plus X, AML Oceanographic, Dartmouth, Canada; internal sensors: conductivity and temperature, CT-Xchange; pressure, Xchange p.x) at 25 m depth at station X (Fig. 1a) between September 2016 and August 2017. Discrete water samples for total alkalinity (TA), dissolved inorganic carbon (DIC), and nutrient concentrations (phosphate, nitrate, silicate) were taken once per season at the coral stations with a Niskin bottle close to the experimental corals (max. distance 2 m). Samples for TA (50 ml) were filtered through glass microfiber filters (GF/F, 0.7 μm pore size; Whatman, GF Healthcare Life Sciences, Amersham, United Kingdom) and kept at 4 °C until analysis within the latest 7 days. TA was determined in four replicate Gran titrations with 0.01 M HCl using a TW alpha plus titrator (SI Analytics, Xylem Analytics, New York, USA) and corrected for Dickson standard seawater (batch 102). Samples for DIC measurements (4.5 ml) were sterile filtered through polycarbonate membrane filters (0.2 μm pore size) and poisoned with 2 μl saturated HgCl$_2$ before storing at 4 °C until analysis. DIC was determined in two replicate samples each measured twice using a QuAAtro39 AutoAnalyser with an XY-2 autosampler (Seal Analytical GmbH, Norderstedt, Germany) and the software AACE (version 7.09). NaHCO$_3$ standards were measured for calibration and to correct the measurements for the methodological drift and samples were corrected with Dickson seawater (batch 161). Nutrient samples (50 ml) were sterile filtered through glass microfibre filters (GF/F, 0.2 μm pore size; Whatman, GF Healthcare Life Sciences, Amersham, UK) and immediately frozen (−20 °C) until analyses at the Pontificia Universidad Católica de Valparaíso with an autoanalyzer (Technicon AutoAnalyzer, Seal Analytical Inc., Wisconsin, USA) after Atlas et al.[102]. Seawater carbonate chemistry parameters (pH, $p$CO$_2$, $\Omega_{arag}$, carbonate ion concentrations [CO$_3^{2-}$]) were calculated from the measured TA, DIC and nutrient concentrations at the experimental water temperature, salinity and pressure using the programme CO2SYS[103] with the dissociation constants for carbonic acid in seawater (K$_1$ and K$_2$) of Luecker et al.[104], for hydrogen sulphate of Dickson[105] and for boric acid of Uppström[106] and pH on the total scale.

**Experimental design**. This field study combined an environmental assessment with coral physiological investigations, including both continuous and discrete measurements. A total of six shallow sampling stations were selected at ~20 m water depth, all located on the steep eastern walls of Comau Fjord, spanning a spatial gradient from the mouth to the head of the fjord (Fig. 1a). In addition to the shallow stations, a deep station at E was established at about 300 m depth (Fig. 1a), coinciding with the vertical pH minimum[44,47]. The in situ physiological assessment of *D. dianthus* was initiated in September 2016 and included a year-long investigation of corals at all seven stations and a reciprocal transplantation experiment between four stations. At regular intervals (every three to four months), the same

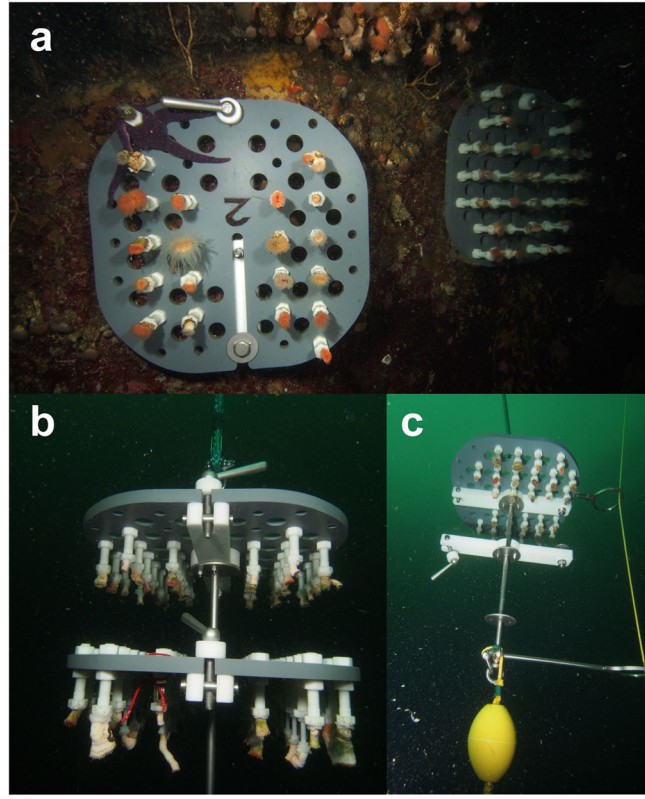

**Fig. 5 *Desmophyllum dianthus* on coral plates and mooring. a** Corals glued on white polyamide screws, which are fixed on grey plastic plates. The plates were fixed on a white plastic holder that was installed on the rock of a coral bank. **b**, **c** Corals on grey plastic plates fixed on a metal rack of mooring. The metal rack was lowered on a pulley (yellow line, **b**) until the yellow stopper buoy reached the anchored stone at 300 m water depth.

coral specimens were collected for measurements of their key metabolic responses in terms of calcification and respiration rates (Fig. 1b).

In the shallow waters of Comau Fjord, corals were collected by scientific SCUBA divers who carefully chiselled the corals from the fjord wall. Corals were collected in ambient water in 1 L closed plastic containers to avoid a potential osmotic shock in the low salinity surface layer while ascending. Care was taken not to expose them to light during transportation on the boat to the research station. The divers drilled holes into the bare rock using a pneumatic drill-hammer (Atlas Copco DKR 36; bit: 10 mm, Atlas Copco, Nacka, Stockholm, Sweden) and inserted and securely fixed plastic holders with stainless-steel bolt anchors (FAZ II 10/30 A4, Fischerwerke GmbH & Co. KG, Waldachtal, Germany) on the fjord wall at the shallow stations to re-install the corals in their natural downward orientation (Fig. 5a). The corals were fixed with polyamide screws on special plastic plates (two to three plates per station, max. 34 corals on each plate; Fig. 5) that were mounted on these holders. The natural density of *D. dianthus* in the fjord was taken into account for the spacing of corals on the plates, but might have been slightly larger than in their natural habitat to facilitate handling and avoid damaging the corals. These plates allowed to re-collect (as well as re-install) subsamples of the corals during each season without disturbing the remaining corals on the other plates. The plates with the corals fitted into black watertight Peli cases (Pelican Products Inc., Torrance, USA) filled with seawater (volume: 80 L) in which they were transported underwater and which protected the corals from the low salinity surface layer as well as from daylight exposure during transport on the boat to the laboratory facilities at the research station. Corals at the deep station were sampled at 280–290 m depth using a remotely operated vehicle (Commander 2, Mariscope Ingeniería, Puerto Montt, Chile; modified with manipulator arms and high-resolution camera) with a wire frame and a bag attached to scrape the corals from the wall and collect them in a 2 m long bag that allowed to insulate the corals on their way through the low salinity surface layer. We do not expect that the corals were differently affected by the two collection techniques, especially given the short sampling period compared to the long experimental period in the field. The few corals that were damaged during collection were not used for the transplantation experiment. On the boat, corals were transferred to a cool box (volume: 120 L) filled with seawater for transportation to the research station. At the deep station, the two coral plates were first installed at the holders of a metal rack at 20 m water depth by scientific SCUBA divers transported in the Peli case. The metal rack was

attached to a pulley and subsequently lowered down to 300 m (Fig. 5b, c). For the re-collection of corals at the deep station, the metal rack with the coral plates was pulled up until 20 m water depth, where the coral plates were removed by divers and transferred into Peli cases for transportation. After the measurements, coral plates were re-installed according to the installation procedure described.

Following the initial collection, all corals were maintained in 20–30 L flow-through aquaria filled with natural seawater pumped from 20 m water depth in front of the research station. The phenotypes of D. dianthus differed between the source populations in shallow and deep waters of the fjord. Some shallow corals (mainly at stations B, C, and Es) were not fully covered with tissue and infested with endolithic photoautotrophic organisms[93,94], whereas corals at 300 m depth were often completely covered with tissue and therefore showed less signs of infestation (Supplementary Fig. 4). In the laboratory, the bare skeletal parts were removed as far as possible using a submerged grinding disc attached to a rotary tool (Dremel 4000, Dremel, Breda, The Netherlands) following Rossbach et al.[23] in order to reduce the skeletal parts of shallow corals affected by bioerosion[93,94]. Care was taken not to damage the coral tissue inside the coelenteron. Afterwards, corals were glued on individually labelled polyamide screws (Toolcraft AG, Georgensgmünd, Germany) using underwater easy glue (Preis Aquaristik KG, Bayerfeld, Germany). Once corals were fixed, handling was only done by touching the screws to prevent any disturbance by direct contact with the corals. The maximum time for maintenance in the laboratory facilities was 3 weeks before the corals were returned to the fjord. Right before the first re-installation, the corals were stained with 50 mg l$^{-1}$ Calcein for 16–19 h in order to mark the beginning of the experiment in the skeleton for further skeletal analyses that are not part of this study. Corals were either re-installed at their collection station (native) or cross-transplanted between stations (novel) where we expected the strongest differences in environmental conditions (Fig. 1), i.e. stations A vs. F (head vs. mouth; horizontal gradient) and stations Es vs. Ed (shallow vs. deep; vertical gradient).

**Coral physiology**. In order to investigate the physiological conditions of the corals as well as seasonal adjustments along the two environmental gradients (horizontal and vertical), a total of 392 corals were collected. Corals from each station were divided in two different sets of corals for the following purposes: (A) A subset was used to monitor their seasonal changes and repeatedly measure their response to changes in their natural environment (experimental corals) and (B) another subset was collected seasonally to measure their biomass (tissue corals; see Supplementary Methods). The experimental corals were used to assess seasonal physiological adjustments by repeatedly measuring their response to changes in their natural environment. While experimental corals were returned to the field, the tissue corals were directly processed for further analysis.

Ten D. dianthus individuals were collected at each of the six shallow stations and 8 individuals at the deep station in September 2016 and fixed on screws (see above). For cross-transplantation, another 10 corals were sampled at shallow stations A, F and Es and 8 corals at the deep station Ed. This gives a total of 68 native corals along the environmental gradient and at depth and 38 novel corals at shallow and deep stations (experimental corals). Overall, only individuals of similar size were sampled in order to reduce the size bias in calcification rates (oral diameter: 18.10 ± 4.47 mm).

Calcification and respiration rates were determined seasonally after 4, 8 and 11 months in austral summer (January 2017), autumn (May 2017) and winter (August 2017), respectively, using the same experimental corals for each station. For this purpose, corals were transported to the research station, unscrewed from the plates and carefully cleaned with a soft toothbrush to remove attached fouling organisms on the screws and bare skeletons. Corals were in the laboratory for a maximum of 7 days before re-exposure.

Respiration rates were determined through closed-cell incubations immediately after re-collection of the corals without acclimatisation to laboratory conditions. The corals were screwed into the lid of 800 ml Schott vials filled to the brim with 100 μm filtered seawater (in order to remove larger plankton organisms and particles) and the vials were closed hermetically. Two Schott vials without corals were used as seawater controls to measure background plankton oxygen consumption. Magnetic stir bars were put into all vials to provide water circulation. The vials were placed in a temperature-controlled water bath (temperature: 12.75 ± 1.44 °C, salinity: 32.1 ± 0.3) on a multi-position magnetic stirrer (MIX 15, 2mag AG, München Germany; stirring rate: 170 rpm). Due to logistical reasons, the temperature of the water bath was controlled by a temperature-controlled room and small temperature differences (<0.5 °C) between shallow stations were not taken into account as they are within the error of the thermostat. We also did not take into account the larger temperature differences between shallow and deep (up to 2 °C in summer and autumn) as we preferred to use uniform temperatures that deviate from in situ temperatures for reasons of standardisation and due to logistical reasons. Therefore, respiration measurements were conducted at standardised conditions, approximately representing water temperatures of shallow stations at the end of each season and seasonally increasing from 11.76 ± 1.63 °C in austral winter, 12.19 ± 0.63 °C in autumn and 14.16 ± 0.43 °C in summer (with more variable temperature conditions in winter as the temperature control system was not working well). This resulted in a seasonally standardised assessment of metabolic rates, which is not related to in situ temperature. Due to this standardisation and because we do not know the temperature performance curves

of D. dianthus, the temperature change between in situ and incubation temperature can lead to either an under- or overestimation of the respiration rates of deep corals. Nevertheless, it allows a direct comparison of the metabolic potential of corals between stations. Corals were incubated for 6 h in the dark. Oxygen (HQ40D multimeter with LDO-101 sensor, Hach Lange GmbH, Düsseldorf, Deutschland), pH (WTW pH 3310, Xylem Analytics, New York, USA) and salinity (WTW cond 3110, Xylem Analytics, New York, USA) were measured prior to the incubations. The oxygen probe was calibrated using 100 % air-saturated water and the pH sensor was calibrated using pH buffers 4, 7 and 10 (WTW, Xylem Analytics, New York, USA) prior to measurements. Oxygen consumption ($\Delta O_2$) was derived from the difference in $O_2$ concentration of the seawater at the beginning and end of the incubations after the background plankton respiration measured in the control vials was subtracted. Respiration rates were normalised to the tissue covered surface area of the corals (see below) and expressed as daily rates (μmol cm$^{-2}$ d$^{-1}$). However, respiration rates were only measured once and at the end of each season and unlike calcification rates, they do not represent the performance of the corals over the entire season.

Initial coral mass (with and without screws and glue) was determined in September 2016 using the buoyant weighing technique[107]. Subsequently, calcification rates were assessed seasonally by measuring buoyant weight one day after re-collection and respiration measurements of the corals. For each coral individual, the buoyant weight was determined with a precision balance (Sartorius CPA 225D-OCE, Sartorius AG, Göttingen, Germany; precision: 0.01 mg) mounted on a platform above a small aquarium filled with seawater from the fjord. Corals were weighed in a metal weighing basket attached to the underfloor unit of the balance and after allowing the corals to acclimate for 15 min to the water temperature and salinity in the small aquarium. Water temperature (ama-digit ad 15th, Amarell GmbH & Co. KG, Kreuzwertheim, Germany) and salinity (WTW cond 3110, Xylem Analytics, New York, USA) were recorded for the subsequent calculation of seawater density and skeletal dry mass after Jokiel et al.[107].

Skeletal aragonite density for D. dianthus (2.793 ± 0.026 g cm$^{-3}$) was derived from 9 experimental corals from stations A and F with overlapping density values after Davies[108] with some slight modifications. For this purpose, the coral tissue was separated from the skeleton using an airbrush (Starter Class set, Revell GmbH, Bünde, Germany) connected to pressurised air at 5 bar and afterwards, bare skeletons were bleached in a 6% sodium hypochlorite solution for 48 h, changing the solution once after 24 h. The corals were split lengthwise and one-half of the skeleton of each specimen was used for measurements of the skeletal density. The buoyant weight of the skeleton in seawater was determined 3 times before rinsing the skeleton with reverse osmotic water (conductivity: 18.0 MΩcm; Sartorius arium pro, Sartorius AG, Göttingen, Germany) and drying to constant mass at 60 °C for about 3 weeks.

Seasonal calcification rates were calculated as the difference between skeletal dry mass at the beginning and end of austral summer (September 2016–January 2017), autumn (January 2017–May 2017) and winter (May 2017–August 2017) per tissue covered coral surface area (see below) and expressed per day (mg cm$^{-2}$ d$^{-1}$) using the following equation:

$$G\left(mg\ CaCO_3\ cm^{-2}d^{-1}\right) = \frac{(M_{t+1} - M_t) \times 1000}{A_{coral} \times t} \qquad (1)$$

where $M_t$ and $M_{t+1}$ are the skeletal dry mass (g) of the specimen at the beginning and the end of each growth period, t is the exposure time in days (d) and $A_{coral}$ is the tissue covered surface area of the coral (cm$^2$, see below). For comparability, calcification rates were additionally normalised to the initial skeletal dry mass at the beginning of each growth period after Orejas et al.[109] and expressed in % d$^{-1}$ (Supplementary Data 2 and Supplementary Fig. 8). This comparison shows that it does not make much difference which reference variable is used (at least if the influence of the bare skeleton that is not covered with tissue is reduced as in the present study).

The tissue covered surface area of the experimental corals was used as a reference variable for calcification (mg cm$^{-2}$ d$^{-1}$) and respiration rates (μmol cm$^{-2}$ d$^{-1}$). The outer tissue covered surface area of all experimental corals was measured at the end of the field study (August 2017) using a digital calliper (reading to 0.01 mm). A detailed description of the surface area measurements with a calliper can be found in the Supplementary Methods (Supplementary Figs. 9, 10). In brief, a modulated formula for a truncated cone following the geometric approximation "Advanced Geometry" by Naumann et al.[110] was used to calculate the inner and outer surface area of the calyx based on the trumpet shape of D. dianthus. For this, the shape of the coral was approximated to a cup and the surface areas of the individual septa were not considered. In addition, the surface area of native and novel corals at station Ed was calculated at the end of each sampling period (January, May and August 2017) using scaled pictures of the corals with the software ImageJ (version 1.52) as the surface areas of the corals at the deep station increased largely during the year of this study (Supplementary Figs. 4, 5). Therefore, the change in tissue covered surface area was considered for the calculation of calcification and respiration rates of the deep corals. Based on the scaled images, the tissue covered surface area of the corals at shallow stations changed only slightly and not substantially (Supplementary Fig. 5). Therefore, the more accurate measurements at the end of the experiment were used for every sampling period for the shallow corals. This difference in surface area determination between shallow and deep corals did not have an essential effect on the calcification results as shown by the comparison with the calcification data

normalised to the skeletal dry mass (Supplementary Fig. 8). The change in tissue covered surface area of the shallow corals over the experimental time was small and the calcification rate at the beginning of the study was therefore only slightly underestimated by using the slightly larger surface area in August. In contrast, the tissue covered surface area of the deep corals was determined retrospectively in each season to minimise the larger error, as the calcification rates in January and May would otherwise have been underestimated.

**Statistics and reproducibility**. All statistical analyses were performed using the software R (version 4.1.0)[111]. As calcification and respiration data were not normally distributed (Shapiro–Wilk test), we used a linear mixed effect model (LMM; *lmer*) to examine the relationship between the response variables calcification and respiration with depth, season, station and transplantation using the R package *lme4*. For this, season, station and station*transplant were considered as fixed factors and coral specimens as a random factor for a repeated measures design. One model was only run with the native corals of the shallow stations to identify changes along the horizontal gradient and between seasons. For a second model, only data of native and novel corals from stations A, F, Es and Ed were used to identify differences between native and novel corals and between depths. Post-hoc comparisons of significant effects were tested using the *lsmeans* function of the package *lsmeans*.

Long-term temperature records for each station were used to quantify the mean temperature variability for the entire year and the different seasons. To do so, temperature data were decomposed into diurnal temperature anomaly values and mean seasonal or annual variability was used as a measure of station-specific environmental variability proxy for other co-varying environmental factors such as seawater pH$_T$, $\Omega_{arag}$, salinity and oxygen concentration. We used the linear model function (*lm*) in R for a multifactorial analysis to test for relationships between calcification and these environmental parameters. Mean seasonal temperature variability, temperature, pH$_T$, $\Omega_{arag}$, salinity and oxygen concentration were used as fixed factors and model selection was performed using the Akaike information criterion (AIC), which was calculated using the R package *AICcmodavg*. We used AIC to go through all models from single environmental to multiple factors without the inclusion of interactions. We used a linear regression approach here because we expected to be in a range of all environmental variables where calcification is approximately linearly related. We tested for normality of residuals with the Kolmogorov–Smirnov test. Model residuals were plotted and assessed for normal distribution and homoscedasticity using the *ols_test_normality* function of the R package *olsrr*.

**Reporting summary**. Further information on research design is available in the Nature Research Reporting Summary linked to this article.

## Data availability

The datasets generated and analysed during the current study are available in the data repository PANGAEA: https://doi.org/10.1594/PANGAEA.945072.

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

## Acknowledgements

We would like to express our deepest thanks to the scientific SCUBA divers Adrian Gruhn, Anna Hohnheiser, Annika Müller, Aurelia Reichardt, Benedikt Caskie, Felix Butschek, Lea Happel, Maximilian Neffe and Thomas Heran and our engineer Henning Schröder for assistance during the fieldwork for this study. We are also very grateful for the help we received from the staff of the Fundación San Ignacio del Huinay, particularly Ulrich Pörschmann, Stacy Ballyram, Francine Beaujot, Insa Stolz, Aris Thomasberger, Mette Schiønning and Darinka Pecarevic, during the fieldwork in Comau Fjord. We would also like to thank Esther Lüdtke, Ulrike Holtz, Beate Müller, Kathrin Vossen, Maria Jung, Aurelia Reichardt and Lea Happel for laboratory assistance and Alexandra Kler Lago and Irini Kioupidi for analysis of the coral pictures. We are in debt to the people of the AWI scientific workshop who produced the custom-made coral holders and plates. Export permits for *Desmophyllum dianthus* according to the Convention on International Trade in Endangered Species of Wild Fauna and Flora (CITES) were admitted by the Chilean Servicio Nacional de Pesca y Acuicultura (Sernapesca; Permiso CITES Nº 17000001WS, 17000003WS, 17000006WS and 17000009WS) and import permits by the German Federal Agency for Nature Conservation (BfN; E00134/17, E00429/17, E00430/17, E103253/17 and E04897/17). This study was conducted within the bilateral Chilean-German project PACOC (Plankton- and cold-water coral ecology in the Comau Fjord, Chile; CONICYT-BMBF: 20140041 and 01DN15024), CONICYT FONDAP-IDEAL 15150003 as well as PACES II-T1WP6 and Changing Earth—Sustaining our Future: Subtopics 4.2 and 6.1 at AWI.

## Author contributions

G.M.S.-G., J.L. and C.R. designed the study. J.L., V.H., G.F. and H.E.G. contributed background information and helped with the fieldwork. J.L. and G.F. collected the corals. V.H. and J.P.E. provided background data. K.K.B. and G.M.S.-G. conducted the measurements. K.K.B., M.W. and G.M.S.-G. analysed and interpreted the data. M.W. and K.K.B. conducted the statistical analysis and prepared the figures. K.K.B., M.W. and C.R. wrote the first draft and all authors edited the manuscript.

## Funding

## Competing interests

The authors declare no competing interests.
