## [Peer Review File · Communications Biology]

Reviewers' comments:

Reviewer #1 (Remarks to the Author):

Dear Authors,

Very clearly, this is one of the best manuscripts I have read. Congratulations, it was a pleasure review this article!

The study presented here investigated the physiological performance and acclimatization potential of the cold-water coral *Desmophyllum dianthus* to changing in situ environmental conditions. In the Comau Fjord in Chile, the species naturally occurs under strong vertical (depth) and horizontal (along-fjord) environmental gradients (e.g. in temperature, aragonite saturation). The Authors used this opportunity to carry out a transplantation experiment, moving corals from deep to shallow and reciprocally, and between the end and the mouth of the fjord. They found that corals perform 'best' at the deeper site, where conditions are most stable, while shallow corals are impacted by strong environmental fluctuations. All transplanted corals acclimatized to the conditions in their new habitat and even adjusted their morphology, demonstrating a high phenotypic plasticity of this cold-water coral.

The study design is very well-conceived and uses a completely novel approach. For the first time, a transplantation experiment on cold-water corals is carried out in situ on a seasonal time scale, including measurement of growth and physiological performance – each of these points is novel on its own. The year-long experiment certainly involved a high level of expertise, planning and dedication; I was particularly impressed by the ROV work combined with SCUBA diving and physiological measurements. As such, the study provides a shining example for future deep-sea experimental research. Likewise, the statistical analysis and the reporting of results are sound, concise and to the point, ensuring reproducibility. The results are crucial for the field of global change research: While we know (or assume) that (cold-water) corals are vulnerable to global change, it remains poorly understood which environmental conditions are most harmful. This study shows, that it is less the average change of, e.g. temperature, but strong fluctuations. Such findings will greatly help to fine-tune our predictions of habitats that might still be suitable for cold-water corals in the future, and should be taken into account in marine spatial planning.

I have only two minor point that may need to be addressed, (1) concerning the temperature in the laboratory incubations (see comments on lines 186, 339, 558), and (2) the normalisation of rates to tissue coverage at the end of the experiments (see comments on lines 594-595). Both points are explained in the line-to-line comments below. They are a minor concern, but should be briefly elaborated on in Methods and Discussion.

The remaining line-to-line comments should be seen as suggestions to even further improve the understandability of this article – but I do not expect the Authors to follow these in detail. In case you prefer, I have also attached this review as pdf.

Again, thank you very much for this work, I am looking forward to see it published and cited soon.

Kindest regards,
Dr. Sandra Maier

#####

Section & line-by-line comments

Abstract

Line 40-41, suggest to split sentence, e.g. "Despite low aragonite 40 saturation levels, it sustains a dense population of the cold-water coral (CWC) *Desmophyllum dianthus*. This situation provides a rare

opportunity to evaluate CWC fitness trade-offs in response to physico-chemical drivers and their variability.”

Line 42ff: One additional sentence explaining the difference between shallow versus deep and head versus mouth could be helpful.

Introduction

Line 59: which one would be biotic, food availability? Just consider and maybe delete ‘and biotic’

Line 61: what is meant by ‘geological time scale’? I assume you mean that adaptation takes long, but maybe clarify this in the text.

Line 71-72: First, I did not understand which data are limiting, maybe simplify/turn around the structure of the section lines 69-75

-few studies carried out so far have found that environmental conditions in the habitat of CWCs vary substantially

-this variability has to be taken into account when assessing the ability of CWCs to cope with changing conditions, but very few studies have done that

Line 89-93: Suggest to simplify, for example:

“In order to assess the resilience of CWCs to future changes of the climate, it is important to first understand their physiological performance under present day in situ conditions and environmental variability. Second, to assess natural environmental variability, long-term studies of oceanographic conditions in situ are of paramount importance.”

Line 93: suggest to delete ‘In addition’ and maybe change to “species living under environmental gradients...”?

Line 101-105: I am not sure if I understand this; are (1) to (3) the possible outcomes of the transplantation experiment, indicating either adaptation, acclimatization or local adaptation? Please clarify, I think this is important to understand the study. For example, phrase as “(1) adaption is indicated if ..., (2) acclimatization if ... and (2) local adaptation if...”

Or “...suggests adaptation (1), ... acclimatization (2), and ...local adaptation...”

Line 116-118: I would here briefly mention what these gradients are; maybe move lines 444-449 from the Methods section up here. That may help the reader follow the Results.

Figure 1:

I really like the figure to explain the experimental design, which is actually quite complicated, but does become quite clear. Maybe you could rephrase the figure legend a little bit, to improve understandability, e.g. (if I understood correctly):

-corals were transplanted as indicated by the colors

-after fixed time points, a subset of corals was sampled for analysis of tissue parameters, indicated by ‘x’ (n=... per station and sampling event)

-in addition, a different subset of corals was used to measure respiration and growth over the entire experimental period, in specific after 4, 8 and 11 months corals, indicated by the rectangles (n=... per station and measurement event).

-Were these corals sampled after 11 months?

Line 132: what is meant by ‘novel corals’, ‘transplanted corals’? I think it would be easier to understand for your Readers if ‘novel’ corals would be called ‘transplanted’ or ‘cross-transplanted’ corals throughout the manuscript.

Methods

Line 451: 'emerges' sounds like a very active process, I'd suggest to rephrase, e.g. to 'In Comau Fjord, the cosmopolitan, azooxanthellate CWC species *Desmophyllum dianthus*, typically a deep-sea species, is found at exceptionally shallow depths of up to 15 m 87,83.'

Line 453-454: 'Chilean fjord region' and 'fjords of North Patagonia' appears redundant, suggest to delete one of these

Line 479 and above: the sampled corals, were they all growing in downward orientation? Or did some corals 'change' their growth direction when transplanted? As I assume this would affect their feeding and hence their physiology?

Could you also remark the fact about their downward growth direction in the text above, e.g. line 465, as this growth 'type' is quite typical of this species (versus other scleractinian CWCs) and potentially even more pronounced in Comau fjord than elsewhere in the world, e.g. the Mediterranean?

Line 481: can you say sth about the spacing of the corals, and if that corresponds to natural conditions? Sure it did, looking at Fig. 5. But maybe mention that you did take individual density into account, because again, this might affect feeding rates and physiology.

Line 494: sounds really adventurous :)

Line 487: do you think the different collection of ROV versus SCUBA divers (probably more careful) affected the results of the study? Given the long period of a year, probably not, right? Not sure if that should be mentioned or not, it's totally up to you.

Line 510: what was the selected size class? Was that mentioned above? If yes, please apologize, if not, maybe add?

Line 510ff: Split sentence. "...of the fjord. Some shallow corals..."

Line 524: suggest to rephrase for readers that are not familiar with the method, e.g. "The calcein incorporated in the coral skeleton marks the start of the experimental growth assessment (see below)." Also I did not find any Results on calcein, or did I miss it? If not reported, I suggest to just delete it entirely.

Line 526: To clarify this and the following, I advise to rephrase to, e.g., "Corals were either re-installed at their collection station (native) or cross-transplanted between stations (transplanted) between the sites where we expected the strongest differences in environmental conditions (Fig. 1), i.e. stations A versus F (mouth versus end; horizontal gradient) and stations Es vs. Ed (shallow versus deep; vertical gradient)."

Line 531: They = 'corals from each station'?

Line 544-546: I would already mention this above, see comment on line 510.

Line 555: 'larger organisms' = 'larger plankton'?

Line 556: suggest 'seawater control measurements' instead of 'control measurements' and mention what these were used for (measure plankton oxygen consumption in the incubation water?)

Line 558: As mentioned in the Review summary: I assume this corresponds to in situ temperatures? But temperature differences between stations (and seasons) were not taken into account in the incubations? Please clarify here and discuss below, see comment on line 339; I know it is very tricky to maintain a precise temperature, but maybe mention this.

Line 567: background plankton respiration

Line 568: related = normalized?

Lines 570: 'expressed as daily rates.'

Line 582: but these were not the experimental specimens, right? Please clarify.

Line 594-595: briefly mention that "the tissue covered coral surface area did not change over the seasons (see below)". But how about the corals transplanted to the deep site, that grew a lot of tissue? Here, you did take changes into account, right, as indicated in lines 611 ff. Please clarify already in this paragraph, or mention that not all rates were normalized to the end value of tissue-covered surface area.

In addition, is this normalization necessary at all? I mean, if you express the values as % of start weight, does the normalization make a difference? Why did you decide to present the data in mg cm⁻² d⁻¹ instead of %?

Line 649: are these half corals the ones used for the density measurement?

Line 654: maybe rephrase 'taken' by 'collected'

Line 714: is it really multivariate, i.e. multiple measurements on the same experimental unit?

Results

Line 186: but these corals were brought from colder into warmer water (in incubations), which might have increased their respiration rates, right? See previous and following comments for methods and discussion (lines 558, 339)

Line 193: this is very exciting, but it was not discussed. Could you add a little bit of info/speculation to this observation in the Discussion, e.g. where you discuss somatic growth? A change in skeletal morphology, and hence overall morphology, wouldn't that be considered a form of adaptation? Why do you think is this trumpet shape advantageous in deep water, does it relate to differences in hydrodynamic conditions/food supply? For example, *Lophelia pertusa* (*Desmophyllum pertusum*) grows different colony shapes depending on the hydrodynamic conditions (e.g. De Clippele et al., 2017). Or do you think it is rather due to physiological differences when calcifying under low aragonite saturation? Or something else? I think you are allowed to speculate a bit in the Discussion, even if you cannot find many references.

DE CLIPPELE, L.H., HUVENNE, V.A.I., OREJAS, C., LUNDÄLV, T., FOX, A., HENNIGE, S.J. & ROBERTS, J.M. (2017) The effect of local hydrodynamics on the spatial extent and morphology of cold-water coral habitats at Tisler Reef, Norway. *Coral Reefs*, 1–14.

Discussion

Line 261-262: Maybe add in brackets what the parameters were for physiological performance and fitness.

Line 263: curious which species these were, you could add this in brackets? So you think all CWCs are that 'fit' in 'acidic' water?

Line 299-300: again I am curious about which species show the positive, no or negative effect; not sure if it is that simple, but if yes, you may add it in brackets (so that the reader does not first have to find the reference)? Or given that the following text nicely summarizes literature, you could consider adding a small table with the respective studies, species, locations, long-versus short term, and effect

on growth.

Line 300: Similar = Similarly?

Line 323-324: what would be a direct, what an indirect effect? Are you referring to the following paragraph?

Line 328: Interesting, is there any explanation why they feed less under high temperature?

Line 339 ff.: Again, these 'deep' corals went from just above 11 degC to almost 13degC; don't you think this increased their respiration rates, at least short-term? So could higher respiration rates be an experimental bias? And may explain the 'deep' corals, though exposed to less food, showed a higher respiration rate in the incubations? I might be wrong or missing something, though. And I don't feel that the respiration data have to be removed, or cannot be discussed as they are. But I do feel this point has to be raised in the Discussion.

Line 339ff: It would be interesting to check if the deep corals also have a higher reproductive output. Also, I find it interesting that the deep corals show an opposite trend in seasonal growth (lower in summer) than the shallow corals (lower in winter). I was wondering previously, if seasonal growth trends might relate to the reproductive cycle, as we speculated for *L. pertusa* (your reference 23). According to Feehan et al. 2019, *D. dianthus* in the region spawns in August and starts gamete production in September; this might be energetically-costly and cause a lower skeletal growth. But the opposite growth trend in deep versus shallow corals seems to contradict this. Not sure if these thoughts are relevant for this article, but maybe relevant for your future research.

Feehan, K. A., Waller, R. G. & Häussermann, V. Highly seasonal reproduction in deep-water emergent *Desmophyllum dianthus* (Scleractinia: Caryophylliidae) from the Northern Patagonian Fjords. *Mar Biol* 166, 52 (2019).

Line 404-405: So is it possible that the corals transplanted from deep to shallow just still had more energy reserves left after a year so that they could afford keeping a higher tissue cover? Or do you have any other speculation why the shallow-origin corals could grow more tissue at the deep site, why can they not do that at the shallow site? Any speculation?

Conclusions

Line 437: environmental variability is increasing in the future, could you provide a reference?

Reviewer #2 (Remarks to the Author):

Article: Environmental stability and phenotypic plasticity benefit the cold-water coral *Desmophyllum dianthus* in an acidified fjord; from Beck et al.

Summary: This study presents the result of transplanting corals along two natural gradients (shallow vs. depth and entrance to further into the fjord) on their fitness. Given low differences due to the origin of the corals, the authors conclude that the corals of this fjord exhibit high phenotypic plasticity.

General comments:

This manuscript addresses a complex and important issue that will help further understand the risk to see cold water coral population disappear with global change. It is generally well-written and easy to understand. It is original and of high interest to the community of cold water coral researchers and broader biology community.

While this work is convincing and represents an impressive novel dataset, I have several questions that I did not find answers to from reading this manuscript:

- 1) Is it possible to claim that the fjord offers a natural gradient? Are conditions in the different sites really different from one another? I do agree with the difference between shallow and depth stations (lines 154-161), but it seems that it might sometimes be misleading to call the head to mouth gradient a natural gradient without proofs that it actually is a gradient? (for eg. In Figure 2, we don't see this gradient).
- 2) Why is there no inclusion of season in the LMM for respiration? (and the second model for growth). Clearly, the season is an important factor to take into account in the model. Or do the authors think that temperature is stable enough to have no seasonal effect? (Seasons are likely operating at least on food quality/quantity, if not directly on metabolic rates?).
- 3) Similarly, why was there no "model 1" for shallow stations and respiration?
- 4) It is difficult (with the material and methods at the end) to not get confused by the growth measurements, I wondered several times what was used in the manuscript. There are information mentioned in the material and method such as calcified growth or calcein staining that are not presented in the manuscript. I would advise the authors to reduce the material and methods by deleting all that information not presented in the results (e.g. lines 592-603, lines 523-524) and move it into supplementary.
- 5) A naïve question but can it be that the generally less fit individuals from B and C are affected by the nearby town activities? (e.g. sewage etc.).

Additionally, I have some minor comments on this manuscript, principally interrogations regarding some parts of the results and their description in the material and methods.

Minor comments:

The introduction is well-written and lays the problematic clearly; it is good that terms such as acclimatization are clearly defined.

- Results:

Figure 2B-E: I would have preferred to see each of the shallow stations (A-F) blue lines, as SD seems to be quite large (for pH for example – see Table S1). And maybe some of the shallow stations are closer to the deep station? It would also make the claim that it is a "gradient" better supported. From Table S1, we can see that station A is close to station Ed for pH. As it is a paper on variability, I would present the variability more clearly. It would also make clear that there is one measurement per station for each season, which should be clearer in the legend (since the material and methods are in the end).

Line 186: This 1st time use of "LMM" is a bit confusing, especially since it does not correspond to any acronym used in the material and methods either.

Line 189: The use of winter/summer can be confusing at first for people from the Northern Hemisphere, maybe it would be good that authors precise that the first time, e.g., "... in winter (august) compared to summer (January) and autumn (may)..."

Line 202: I personally believe that a figure similar to Figure S6 with temperature would be useful data for other scientists to be able to compare their data with (not just temperature variability).

Line 199: The p-value presented here is not the same as the one in Table S3. Is this a mistake? Or did I misread the results?

- Discussion:

Line 255: If I understood correctly, the "fittest" individuals are defined as the fastest growing? I would add this in e.g. '...found the fittest (i.e. fastest growing) D. dianthus...' This is not the only definition of

fittest and the reproductive output is also part of this equation, it seems that the fastest-growing individuals are also the individuals that have the highest metabolic rate, and if this was a consequence of a stress (not sure it is the case here), it could prevent individuals to invest in reproductive growth. I would be actually interested in the ratio between respiration and somatic growth (how much respiration is needed to grow 1 mg: $\mu\text{mol O}_2$ per mg^{-1} – for each individual), as it also relates to the discussion on lines 320-324. I made a quick figure from the means provided and it looks like these ratios differ widely by station and by season (see figure attached). If this turns out to be interesting, I would recommend that it is added as a Figure 3C, it correlates with the result that B and C are in “trouble” (line 383) with high respiration rates for minimal growth rates.

While not a needed addition to the discussion, the authors could be interested in a recent paper from Chilean environmental data (and variability) by Vargas et al. from 2022 in Nature Climate Change (12, 200-207).

- Material and methods:

Line 483...: Was the transport temperature-controlled? Was the temperature the same as the sampling site? (also lines 522)

Line 559: It needs to be precised in the legend of Fig 3B that it is respiration rates at 12.5°C, it is not “true” respiration on site and in the site conditions. Why was respiration rates measured at 12°C and not at the temperature of the sampling site? Do we expect no difference due to 2-3°C differences? How many days after collection were the respiration trials done? Had they acclimatized to laboratory conditions? Were the corals starved before starting the trials? All these are critical information for other scientists to be able to compare their results.

Line 582: Was there a difference with different stations in skeletal aragonite density? If this was not tested ($n=9$), which stations were these specimens from?

Line 662: I believe that the Environmental data paragraph should come right after the field site description, as it is the way the results are presented.

Line 696-706: As said above, I have trouble understanding why the results of the first model are not presented for respiration in the manuscript? And why seasons are not included as a fixed factor for the second model?

Line 711-716: I am not sure that I understand this part, is this single factor linear regressions (e.g. $\text{lm}(\text{growth} \sim \text{temperature})$ or $\text{lm}(\text{growth} \sim \text{temperature} + \text{salinity})$ and then it is compared by AIC)? I think this calls for a little more explanation on why choose these methods and what it means. Are the factors really linearly correlated? Maybe if these figures would be presented in the supplementary material, it would be easier to understand.

Reviewer #3 (Remarks to the Author):

Dear editor,
Dear authors,

I reviewed with great pleasure the manuscript “Environmental stability and phenotypic plasticity benefit the cold-water coral *Desmophyllum dianthus* in an acidified fjord” of K.K. Beck and co-workers. This submitted paper presents a very detailed and well described study about the cold-water coral species *Desmophyllum dianthus*, which had been transplanted to several sites and water depths to test coral’s acclimatisation to novel environments and their development also under variable environmental settings (temperature, aragonite saturation states, salinity, pH). These data and the study are showing first results of the CWC’s ability to react on changed environmental parameters in a fjord system and are new and very important in understanding possible reaction of CWCs to future scenarios. The data presented and the methodology is described in very detail and transparent enabling the reproducibility of this study. The conclusions are clearly presented, but I would add aspects on seawater carbonate chemistry and calcification / growth of the

coral to complete the picture of coral's fitness and adaption (see below).

Overall, the manuscript is of very good quality, language, figures and data presentation and suits very well to the journal's scope. Therefore, I recommend publication with only very minor revisions:

1) I indicated in the annotated pdf, where I would include additional citation of references, which might have been overlooked, but I believe should be integrated to complete the picture and acknowledge these previous works.

2) I understand well that the authors focus mainly on the temperature variability in their study, as a strong correlation between temperature fluctuations and other environmental parameters has been observed (4,7,24,25,48). Nevertheless, I would like to see additional discussion about the similar negative correlation of pH or aragonite saturation state with growth (figure S6) especially having the "acidified fjord" in the title of the study. This comparable correlation exists for the months of May and August - but not January. Why is this the case? Is that a result of large T-variability in January compared to May and August? Could this be related to better (winter) mixing in January? Both parameters (or the whole parameters of seawater carbonate chemistry) are indispensable for CWCs during the calcification process as e.g., *Desmophyllum pertusum* mainly relies on an external DIC source for the calcification process (Mueller et al., 2014). Further, it had been suggested that CWCs can increase their internal DIC concentration (McCulloch et al., 2012), but strongly elevated seawater DIC can possibly not be regulated by metabolic/calcification processes (Flögel et al., 2014) - a closer look to the CO₂ system is therefore essential and should be included while considering fitness, growth or respiration of CWCs.

As the authors present this comprehensive data set including all seawater carbonate chemistry parameters, I would recommend not to omit these aspects important for CWC's adaptability and include them into the discussion (as additional paragraph) and conclusion.

Thanks a lot and kind regards!

References:

Flögel, S., Dullo, W.-Chr., Pfannkuche, O., Kiriakoulakis, K., Rüggeberg, A. (2014) Geochemical and physical constraints for the occurrence of living cold-water corals. *Deep-Sea Research II* 99: 19–26, doi: 10.1016/j.dsr2.2013.06.006.

McCulloch, M., Trotter, J., Montagna, P., Falter, J., Dunbar, R., Freiwald, A., Försterra, G., López Correa, M., Maier, C., Rüggeberg, A., Taviani, M. (2012) Resilience of cold-water scleractinian corals to ocean acidification: Boron isotopic systematics of pH and saturation state up-regulation. *Geochimica et Cosmochimica Acta* 87:21-34, doi:10.1016/j.gca.2012.03.027.

Mueller, C.E., Larsson, A.I., Veuger, B., Middelburg, J.J., van Oevelen, D. (2014) Opportunistic feeding on various organic food sources by the cold-water coral *Lophelia pertusa*. *Biogeosciences* 11:123-133. Doi:10.5194/bg-11-123-2014

Dear Editor, dear Reviewers,

First we want to thank you for expressing your interest in our study and the opportunity to prepare and submit a revised version of our manuscript. We are grateful for the very positive reviewer responses and their detailed and extremely helpful comments. We have now incorporated all suggestions into our revised manuscript and provide detailed responses below.

Reviewer #1 :

Dear Authors,

Very clearly, this is one of the best manuscripts I have read. Congratulations, it was a pleasure review this article! [...] This study shows, that it is less the average change of, e.g. temperature, but strong fluctuations. Such findings will greatly help to fine-tune our predictions of habitats that might still be suitable for cold-water corals in the future, and should be taken into account in marine spatial planning.

R: We thank Sandra Maier for her very positive assessment and we are happy that she enjoyed reading our manuscript.

Line 40-41, suggest to split sentence, e.g. “Despite low aragonite 40 saturation levels, it sustains a dense population of the cold-water coral (CWC) *Desmophyllum dianthus*. This situation provides a rare opportunity to evaluate CWC fitness trade-offs in response to physico-chemical drivers and their variability.”

R: Modified accordingly.

Line 42ff: One additional sentence explaining the difference between shallow versus deep and head versus mouth could be helpful.

R: We understand this suggestion, but due to the limit of 150 words, we are already beyond this limit and find it difficult to add this aspect. Thus, we decided to not include it in the abstract.

Line 59: which one would be biotic, food availability? Just consider and maybe delete ‘and biotic’

R: We deleted “abiotic and biotic factors”.

Line 61: what is meant by 'geological time scale'? I assume you mean that adaptation takes long, but maybe clarify this in the text.

R: Yes, we meant that adaptation takes long, but we deleted "geological time scales" now to avoid confusion.

Line 71-72: First, I did not understand which data are limiting, maybe simplify/turn around the structure of the section lines 69-75

-few studies carried out so far have found that environmental conditions in the habitat of CWCs vary substantially

-this variability has to be taken into account when assessing the ability of CWCs to cope with changing conditions, but very few studies have done that

R: Modified accordingly.

Line 89-93: Suggest to simplify, for example:

"In order to assess the resilience of CWCs to future climate changes, it is important to first understand their physiological performance under present day *in situ* conditions and environmental variability. Second, to assess natural environmental variability, long-term studies of oceanographic conditions *in situ* are of paramount importance."

R: Modified as: "In order to assess the resilience of CWCs to future changes of the climate, it is important to first understand their physiological performance under present day *in situ* conditions and environmental variability. Therefore, long-term studies of oceanographic conditions *in situ* are paramount to assess natural environmental variability."

Line 93: suggest to delete 'In addition' and maybe change to "species living under environmental gradients..."?

R: Modified accordingly.

Line 101-105: I am not sure if I understand this; are (1) to (3) the possible outcomes of the transplantation experiment, indicating either adaptation, acclimatization or local adaptation?

Please clarify, I think this is important to understand the study. For example, phrase as "(1) adaption is indicated if ..., (2) acclimatization if ... and (2) local adaptation if..."

Or "...suggests adaptation (1), ... acclimatization (2), and ...local adaptation..."

R: Yes, 1-3 are the three possible outcomes of the transplantation experiment. We rephrased it and hope that it is clearer now (line 101): "A transplantation experiment can show either (1) adaptation if the organism's performance depends on its origin and is not influenced by the environment, (2) local adaptation if the organism's performance depends on both origin

and environmental conditions, or (3) acclimatization if the organism's performance depends on the environment, regardless of its origin^{39,40,41}."

Line 116-118: I would here briefly mention what these gradients are; maybe move lines 444-449 from the Methods section up here. That may help the reader follow the Results.

R: We added some information on the gradients (line 118): "We took advantage of the occurrence of *D. dianthus* in contrasting environments in the stratified Comau Fjord, with low salinity, high pH and oxygen concentrations in surface waters and marine conditions, but low pH and oxygen levels in deeper waters^{44,45,46,47,48}."

Figure 1:

I really like the figure to explain the experimental design, which is actually quite complicated, but does become quite clear. Maybe you could rephrase the figure legend a little bit, to improve understandability, e.g. (if I understood correctly):

- corals were transplanted as indicated by the colors
- after fixed time points, a subset of corals was sampled for analysis of tissue parameters, indicated by 'x' (n=... per station and sampling event)
- in addition, a different subset of corals was used to measure respiration and growth over the entire experimental period, in specific after 4, 8 and 11 months corals, indicated by the rectangles (n=... per station and measurement event).
- Were these corals sampled after 11 months?

R: Thank you for the suggestions for improvement. We rephrased the figure legend and hope that it is clearer now (line 137): "Figure 1: Experimental design and coral sampling scheme. a) Coral sampling stations in Comau Fjord, Chile: six stations at 20 m water depth (A-F shallow, blue colours) and one station at 300 m water depth (E deep, yellow). The research station in Huinay is located between station B and C (star). The CTD was deployed at 25 m water depth at the station X. b) The experimental design includes vertical and horizontal reciprocal transplantation of novel (i.e. cross-transplanted) corals between the shallow stations A and F as well as between shallow (E shallow: Es) and deep (E deep: Ed), where colours indicate the station of origin. Corals collected at stations B, C and D were only returned to their respective native station. One subset of corals (experimental corals) was used repeatedly for calcification and respiration measurements over the entire experimental period (i.e. after four, eight and eleven months; number of individuals N_e : 8-10 per station and sampling time point). A second subset of corals (tissue corals) was sampled for biomass analysis after four, eight and eleven months (N_t : 6-10 per station and sampling time point; see Supplementary Methods and Results). The tissue corals were initially sampled for the same time points as experimental corals, but due to logistical problems could only be obtained for the time points and stations marked with an X."

Line 132: what is meant by ‘novel corals’, ‘transplanted corals’? I think it would be easier to understand for your Readers if ‘novel’ corals would be called ‘transplanted’ or ‘cross-transplanted’ corals throughout the manuscript.

R: Yes, “novel” corals are the cross-transplanted corals between stations A-F and Es-Ed. We now wrote “novel (i.e. cross-transplanted) corals” in the text when first mentioning “novel corals” and hope that it is clearer now. Because of uniformity we would like to stick to “native” and “novel”, which has also been used by previous reciprocal transplantation studies (e.g. Rocker et al., 2019).

Rocker, M. M., Francis, D. S., Fabricius, K. E., Willis, B. L., & Bay, L. K. Temporal and spatial variation in fatty acid composition in *Acropora tenuis* corals along water quality gradients on the Great Barrier Reef, Australia. *Coral Reefs*, 38, 215-228 (2019).

Methods

Line 451: ‘emerges’ sounds like a very active process, I’d suggest to rephrase, e.g. to ‘In Comau Fjord, the cosmopolitan, azooxanthellate CWC species *Desmophyllum dianthus*, typically a deep-sea species, is found at exceptionally shallow depths of up to 15 m 87,83.’

R: Modified accordingly.

Line 453-454: ‘Chilean fjord region’ and ‘fjords of North Patagonia’ appears redundant, suggest to delete one of these

R: Modified accordingly.

Line 479 and above: the sampled corals, were they all growing in downward orientation? Or did some corals ‘change’ their growth direction when transplanted? As I assume this would affect their feeding and hence their physiology?

Could you also remark the fact about their downward growth direction in the text above, e.g. line 465, as this growth ‘type’ is quite typical of this species (versus other scleractinian CWCs) and potentially even more pronounced in Comau fjord than elsewhere in the world, e.g. the Mediterranean?

R: Yes, the sampled corals were also growing downward as *D. dianthus* is naturally growing in a downward orientation in Comau Fjord and therefore, did not change their growth direction due to transplantation. We added the sentence “In Comau Fjord, *D. dianthus* mainly grows under overhangs, the underside of rocks and on the steep fjord walls with the calyx oriented downward^{103,110,111}, presumably to prevent negative effects of high sedimentation rates in this region.” at the beginning of the method section (line 541).

Line 481: can you say sth about the spacing of the corals, and if that corresponds to natural conditions? Sure it did, looking at Fig. 5. But maybe mention that you did take individual density into account, because again, this might affect feeding rates and physiology.

R: We added this information (line 603): “The natural density of *D. dianthus* in the fjord was taken into account for the spacing of corals on the plates. However, the spacing on the plates might have been slightly larger than in their natural habitat to facilitate handling and avoid damaging the corals.”

Line 494: sounds really adventurous :)

R: Indeed the entire expedition was quite an endeavor and required quite some logistical and experimental efforts.

Line 487: do you think the different collection of ROV versus SCUBA divers (probably more careful) affected the results of the study? Given the long period of a year, probably not, right? Not sure if that should be mentioned or not, it's totally up to you.

R: We do not think that the corals were impacted by any of the two sampling techniques, as you already pointed out and we included the following information in the manuscript (line 616): “We do not expect that the corals were differently affected by the two collection techniques, especially given the short sampling period compared to the long experimental period in the field. The few corals that were damaged during collection were not used for the transplantation experiment.”

Line 510: what was the selected size class? Was that mentioned above? If yes, please apologize, if not, maybe add?

R: The size class was specifically mentioned for the experimental corals in the manuscript already, as we considered it especially relevant for the growth measurements (line 678): “Overall, only individuals of similar size were sampled in order to reduce the size bias in growth rates (oral diameter: 18.10 ± 4.47 mm)”. As we moved all information about the tissue corals, which were used for determination of the biomass, into the supplements now because the biomass data are shown in the supplements, we also added the size class for the tissue corals in the supplements (line 1410): “The size of tissue corals (oral diameter: 14.96 ± 3.94 mm) was representative for the population.”

Line 510ff: Split sentence. “...of the fjord. Some shallow corals...”

R: Modified accordingly.

Line 524: suggest to rephrase for readers that are not familiar with the method, e.g. “The calcein incorporated in the coral skeleton marks the start of the experimental growth assessment (see below).” Also I did not find any Results on calcein, or did I miss it? If not reported, I suggest to just delete it entirely.

R: Calcification rates were only measured using the buoyant weighing technique. However, we think that it is important to mention the calcein staining as it may affect the performance of the corals. We added the following information in the manuscript (line 653): “Right before the first re-installation, the corals were stained with 50 mg l⁻¹ Calcein for 16-19 h in order to mark the beginning of the experiment in the skeleton for further skeletal analyses that are not part of this study.”

Line 526: To clarify this and the following, I advise to rephrase to, e.g., “Corals were either re-installed at their collection station (native) or cross-transplanted between stations (transplanted) between the sites where we expected the strongest differences in environmental conditions (Fig. 1), i.e. stations A versus F (mouth versus end; horizontal gradient) and stations Es vs. Ed (shallow versus deep; vertical gradient).”

R: Modified accordingly. However, we stick to the term “native” and “novel” instead of “native” and “transplanted” as “novel” was defined before for the cross-transplanted corals (see comment to line 132).

Line 531: They = ‘corals from each station’?

R: Modified accordingly.

Line 544-546: I would already mention this above, see comment on line 510.

R: Modified accordingly.

Line 555: ‘larger organisms’ = ‘larger plankton’?

R: Modified accordingly.

Line 556: suggest ‘seawater control measurements’ instead of ‘control measurements’ and mention what these were used for (measure plankton oxygen consumption in the incubation water?)

R: Modified accordingly.

Line 558: As mentioned in the Review summary: I assume this corresponds to *in situ* temperatures? But temperature differences between stations (and seasons) were not taken into account in the incubations? Please clarify here and discuss below, see comment on line 339; I know it is very tricky to maintain a precise temperature, but maybe mention this.

R: We did not take temperature differences between stations into account for the incubations as we preferred to use uniform temperatures that deviate from the *in situ* temperature at each of the stations and seasons for reasons of standardisation and due to logistical reasons. As the temperature of the water bath was only controlled by the temperature-controlled room at the research station, it was not possible to adjust the water temperature to the exact *in situ* temperature of each station. As the horizontal differences between the shallow stations are small (< 0.5 °C) and within the error of the thermostat, differences in *in situ* temperatures between shallow stations were not taken into account. The differences between shallow and deep are small in winter, but can be up to 2 °C in summer and autumn. However, we decided to accept this experimental bias in order to standardise the respiration measurements.

To make this clear we added the following information in the methods (line 697ff): “Due to logistical reasons, the temperature of the water bath was controlled by a temperature-controlled room and small temperature differences (< 0.5 °C) between shallow stations were not taken into account as they are within the error of the thermostat. We also did not take into account the larger temperature differences between shallow and deep (up to 2 °C in summer and autumn) as we preferred to use uniform temperatures that deviate from *in situ* temperatures for reasons of standardisation and due to logistical reasons. Therefore, respiration measurements were conducted at standardised conditions, approximately representing water temperatures of shallow stations at the end of each season and seasonally increasing from 11.76 ± 1.63 °C in austral winter, 12.19 ± 0.63 °C in autumn and 14.16 ± 0.43 °C in summer (with more variable temperature conditions in winter as the temperature control system was not working well). This resulted in a seasonally standardised assessment of metabolic rates, which is not related to *in situ* temperature. Due to this standardisation and because we do not know the temperature performance curves of *D. dianthus*, the temperature change between *in situ* and incubation temperature can lead to either an under- or overestimation of the respiration rates of deep corals. Nevertheless, it allows a direct comparison of the metabolic potential of corals between stations.”

Line 567: background plankton respiration

R: Modified accordingly.

Line 568: related = normalized?

R: Modified accordingly.

Lines 570: 'expressed as daily rates.'

R: Modified accordingly.

Line 582: but these were not the experimental specimens, right? Please clarify.

R: No, the specimens for the determination of the skeletal density were from stations A and F (eight specimens from station A and one specimen from station F). The value of the station F specimen was within the range of values of the 8 station A specimens. We included this information in the manuscript now (line 738): "Skeletal aragonite density for *D. dianthus* ($2.793 \pm 0.026 \text{ g cm}^{-3}$) was derived from nine experimental corals from stations A and F with overlapping density values after Davies¹⁰⁶."

Line 594-595: briefly mention that "the tissue covered coral surface area did not change over the seasons (see below)". But how about the corals transplanted to the deep site, that grew a lot of tissue? Here, you did take changes into account, right, as indicated in lines 611 ff. Please clarify already in this paragraph, or mention that not all rates were normalized to the end value of tissue-covered surface area.

R: Thanks for pointing out that this was confusing. We only mention in this paragraph now that the calcification rates were normalized to tissue covered surface area, but deleted the information that the values at the end of the experiment were used (also in the paragraph about the respiration measurements), as you suggested (line 749): "Seasonal calcification rates were calculated as the difference between skeletal dry mass at the beginning and end of austral summer (September 2016 – January 2017), autumn (January 2017 – May 2017) and winter (May 2017 – August 2017) per tissue covered coral surface area (see below) and expressed per day ($\text{mg cm}^{-2} \text{ d}^{-1}$)."

A detailed description that the change in tissue covered surface area of the deep corals was taken into account is provided in the next paragraph (lines 766 and 774): "The outer tissue covered surface area of all experimental corals was measured at the end of the field study (August 2017) using a digital calliper (reading to 0.01 mm)." and "In addition, the surface area of native and novel corals at station E deep was calculated at the end of each sampling period (January, May and August 2017) using scaled pictures of the corals with the software ImageJ (version 1.52) as the surface areas of the corals at the deep station increased largely during the year of this study (Supplementary Figures 7 and 8). Therefore, the change in tissue covered surface area was considered for the calculation of calcification and respiration rates of the deep corals."

We also added a more detailed explanation for the different approaches for the surface area determination (line 783): "This difference in tissue covered surface area determination between shallow and deep corals did not have an essential effect on the calcification results as shown by the comparison with the calcification data normalised to the skeletal dry mass

(Supplementary Figure 6). The change in tissue covered surface area of the shallow corals over the experimental time was small and the calcification rate at the beginning of the study was therefore only slightly underestimated by using the slightly larger surface area in August. In contrast, the tissue covered surface area of the deep corals was determined retrospectively in each season to minimise the larger error, as the calcification rates in January and May would otherwise have been underestimated.”

In addition, is this normalization necessary at all? I mean, if you express the values as % of start weight, does the normalization make a difference? Why did you decide to present the data in $\text{mg cm}^{-2} \text{d}^{-1}$ instead of %?

R: We decided to use the normalization to tissue covered surface area as calcification is only possible in the part of the skeleton that is covered with tissue. As we removed the bare skeletal parts before the start of the experiment, it was not relevant for the present study and calcification rates did not differ significantly between values expressed in $\% \text{d}^{-1}$ of start weight and in $\text{cm}^{-2} \text{d}^{-1}$ normalized to the tissue covered surface area. However, we think that it makes generally more sense to express calcification rates normalized to the tissue covered surface area as the amount of bare skeletal parts can largely affect the result if calcification rates are normalized to the start weight and the tissue covered surface area largely differs between corals (as described in more detail in the supplementary material, line 1351). In this case, the calcification rate of a large individual with a small tissue covered surface area would be underestimated, whereas the calcification rate of a small individual that is completely covered with tissue would be overestimated. However, we decided to also state the values in $\% \text{d}^{-1}$ in table 2 in the supplements for comparability with other studies.

We also added a graph with the calcification rates in $\% \text{d}^{-1}$ in the supplements now (Supplementary Figure 6) and also added an explanation in the manuscript (line 760): “This comparison shows that it does not make much difference which reference variable is used (at least if the influence of the bare skeleton that is not covered with tissue is reduced as in the present study).”

Line 649: are these half corals the ones used for the density measurement?

R: No, we used corals from stations A and F for the density measurements. We only split the deep corals in halves because we did not have as many deep specimens as shallow ones and wanted to keep a part of these few deep corals for further analyses (which are not included in this manuscript).

Line 654: maybe rephrase ‘taken’ by ‘collected’

R: Modified accordingly.

Line 714: is it really multivariate, i.e. multiple measurements on the same experimental unit?

R: Thanks for pointing this out and we indeed used the wrong term here and changed it to “multifactorial” now as we tested the influence of different factors on a response variable.

Results

Line 186: but these corals were brought from colder into warmer water (in incubations), which might have increased their respiration rates, right? See previous and following comments for methods and discussion (lines 558, 339)

R: Yes, the higher temperatures may lead to elevated respiration rates in deep corals compared to shallow corals. However, as we do not know the temperature performance curves of *D. dianthus*, we can only speculate if the change in temperature leads to an under- or overestimation of the respiration rates of deep corals.

Overall, we aimed to use approximately similar temperature for the respiration measurements across stations and within season (so warmest temperatures in summer and coldest in winter) to compare respiration between stations irrespective of *in situ* temperatures (which was due to logistical reasons - see our response to the point raised in the method section and line 558), which will also change from the beginning to the end of each seasonal period. As we expect that the deep corals have higher mitochondria density/activity (based on increased calcification rates, higher biomass as well as expected enhanced nutrition), this should result in elevated respiration rates under similar temperature. Thus, by using the same temperature we rather provide insights into the metabolic potential than the exact *in situ* rate.

In addition, the effect of temperature change of 2°C from 11.8 °C in winter to 14.2 °C in summer can be seen for instance in the native shallow corals in Es where the respiration rate changed from 1.7 $\mu\text{mol cm}^{-2} \text{d}^{-1}$ in winter to a max of 4.2 $\mu\text{mol cm}^{-2} \text{d}^{-1}$ in summer. This suggests that an approx. 3°C increase (from 11.3 °C *in situ* to 14.2 °C in incubations in summer) could lead to approx. 5 $\mu\text{mol cm}^{-2} \text{d}^{-1}$ but not to 8 $\mu\text{mol cm}^{-2} \text{d}^{-1}$ as measured for the deep corals in summer. The generally twice as high metabolic rate of deep corals over the seasons is therefore rather an indication of a changed metabolism in the deep corals.

We now included more information about this topic in the methods (line 697ff) and in the Supplementary Discussion (line 1597ff).

Line 193: this is very exciting, but it was not discussed. Could you add a little bit of info/speculation to this observation in the Discussion, e.g. where you discuss somatic growth? A change in skeletal morphology, and hence overall morphology, wouldn't that be considered a form of adaptation? Why do you think is this trumpet shape advantageous in deep water, does it relate to differences in hydrodynamic conditions/food supply? For example, *Lophelia pertusa* (*Desmophyllum pertusum*) grows different colony shapes depending on the

hydrodynamic conditions (e.g. De Clippele et al., 2017). Or do you think it is rather due to physiological differences when calcifying under low aragonite saturation? Or something else? I think you are allowed to speculate a bit in the Discussion, even if you cannot find many references.

DE CLIPPELE, L.H., HUVENNE, V.A.I., OREJAS, C., LUNDÄLV, T., FOX, A., HENNIGE, S.J. & ROBERTS, J.M. (2017) The effect of local hydrodynamics on the spatial extent and morphology of cold-water coral habitats at Tisler Reef, Norway. *Coral Reefs*, 1–14.

R: We expect that this change in shape is connected to the hydrodynamic regime at the transplant location at the deep station. It is likely that the hydrodynamic regime also influences the access to food but may not be the main cause for change in morphology. Since the change is not related to the origin of the corals, it is an acclimatisation and response to the given environmental conditions at the deep station on the pulley (likely changed hydrodynamic regime) and affects both deep and shallow corals alike. However, we cannot judge whether this change in shape is advantageous or not since this was not the aim of the study.

We now included this aspect in the discussion (line 429): “The morphological change of the native and novel deep corals on the pulley in the water column is also a sign of acclimatisation to the environmental conditions in deep waters. However, we cannot say to what extent biological factors (e.g. supposedly lower predation) or environmental factors (e.g. stronger currents) may also have played a role compared to the corals on the fjord wall. For example, it was observed that *L. pertusa* colonies grow in different shapes depending on the hydrodynamic conditions¹⁰¹.”

Discussion

Line 261-262: Maybe add in brackets what the parameters were for physiological performance and fitness.

R: We added that the calcification measurements are meant here.

Line 263: curious which species these were, you could add this in brackets? So you think all CWCs are that ‘fit’ in ‘acidic’ water?

R: We added the information that these reefs are mainly dominated by *L. pertusa*. However, also other CWC species may be present, but this information is not provided in the three papers (Thresher et al., 2012; Lunden et al., 2013; Georgian et al., 2016). The main problem here is that most studies focus on *L. pertusa*, *D. dianthus* and *M. oculata*. Even though it has been shown that all of these three species seem to be quite resilient to aragonite undersaturation, we know very little about other species so far, but think that other CWC species will also be as “fit” in “acidic” waters of Comau Fjord as *D. dianthus*. However, other

factors than pH are probably also very important for the fitness of these species in aragonite undersaturated waters, as e.g. range and variability of other environmental factors, enough food availability *in situ* and in many laboratory studies.

Thresher, R. E., Tilbrook, B., Fallon, S., Wilson, N. C. & Adkins, J. Effects of chronic low carbonate saturation levels on the distribution, growth and skeletal chemistry of deep-sea corals and other seamount megabenthos. *Mar. Ecol. Prog. Ser.* 442, 87–99 (2011).

Lunden, J. J., Georgian, S. E. & Cordes, E. E. Aragonite saturation states at cold-water coral reefs structured by *Lophelia pertusa* in the northern Gulf of Mexico. *Limnol. Oceanogr.* 58, 354–362 (2013).

Georgian, S. E. *et al.* Oceanographic patterns and carbonate chemistry in the vicinity of cold-water coral reefs in the Gulf of Mexico: Implications for resilience in a changing ocean. *Limnol. Oceanogr.* 61, 648–665 (2016).

Line 299-300: again I am curious about which species show the positive, no or negative effect; not sure if it is that simple, but if yes, you may add it in brackets (so that the reader does not first have to find the reference)? Or given that the following text nicely summarizes literature, you could consider adding a small table with the respective studies, species, locations, long-versus short term, and effect on growth.

R: We included a table with all studies including the species, location, treatment conditions and duration about the impact of temperature and pH on calcification rates of CWCs (Table 1).

Line 300: Similar = Similarly?

R: We stick to “similar” here as we mean that similar variable results were found here and not “similarly, variable results...”

Line 323-324: what would be a direct, what an indirect effect? Are you referring to the following paragraph?

R: Yes, we are referring to the following paragraph here. As mentioned in the next paragraph, the zooplankton community may also be affected by environmental variability and therefore, affecting the food availability of corals. We adapted this sentence now to make it clearer (line 366): “However, it still needs to be elucidated in more detail whether environmental fluctuations have a direct or only indirect effect on coral fitness, e.g. by influencing their food availability.”

Line 328: Interesting, is there any explanation why they feed less under high temperature?

R: Reduced polyp activity as found by Chapron et al. (2021) likely leads to reduced capture rates. We modified the sentence accordingly (line 369): “Due to high environmental variability in shallow waters in Comau Fjord, corals are likely to feed less during periods of elevated temperatures, as suggested by the low prey capture rates of *Lophelia pertusa* and *Madrepora oculata* in laboratory experiments at higher temperatures (17 °C)⁷⁴, which are probably a consequence of the lower polyp activities found in the same study.”

Chapron, L., Lartaud, F., Bris, N. Le & Galand, P. E. Local Variability in Microbiome Composition and Growth Suggests Habitat Preferences for Two Reef-Building Cold-Water Coral Species. *Front. Microbiol.* 11, 1–11 (2020).

Line 339 ff.: Again, these ‘deep’ corals went from just above 11 degC to almost 13degC; don’t you think this increased their respiration rates, at least short-term? So could higher respiration rates be an experimental bias? And may explain the ‘deep’ corals, though exposed to less food, showed a higher respiration rate in the incubations? I might be wrong or missing something, though. And I don’t feel that the respiration data have to be removed, or cannot be discussed as they are. But I do feel this point has to be raised in the Discussion.

R: Yes, increasing the incubation temperature for deep corals most likely increased their respiration rates during the 6 hours of measurement, but we accepted this experimental bias because we wanted to standardise the respiration measurements for better comparison of respiration rates between stations (for details see comment above to result line 186). To better understand the respiration data we would have needed additional measurements (e.g. similar seasonally standardized temperature as used in the present study vs. *in situ* temperature, time for starvation to limit the contribution of digestion to the respiration rate that may differ between sites etc.).

Line 339ff: It would be interesting to check if the deep corals also have a higher reproductive output. Also, I find it interesting that the deep corals show an opposite trend in seasonal growth (lower in summer) than the shallow corals (lower in winter). I was wondering previously, if seasonal growth trends might relate to the reproductive cycle, as we speculated for *L. pertusa* (your reference 23). According to Feehan et al. 2019, *D. dianthus* in the region spawns in August and starts gamete production in September; this might be energetically-costly and cause a lower skeletal growth. But the opposite growth trend in deep versus shallow corals seems to contradict this. Not sure if these thoughts are relevant for this article, but maybe relevant for your future research.

Feehan, K. A., Waller, R. G. & Häussermann, V. Highly seasonal reproduction in deep-water emergent *Desmophyllum dianthus* (Scleractinia: Caryophylliidae) from the Northern Patagonian Fjords. *Mar Biol* 166, 52 (2019).

R: Yes, it would definitely be interesting to study the reproductive cycle of shallow and deep coral of Comau Fjord in future studies. We also included this aspect in the discussion now (line 475ff): “In contrast, somatic growth is clearly sacrificed at the shallow stations, maybe to maintain reproductive output¹⁰⁶. As reproduction requires substantial energy²⁵, low tissue coverage and calcification rates of shallow corals could potentially indicate an energetic trade-off. In addition, decreasing calcification rates may not or not solely be linked to decreased seasonal temperatures, but indicate that more energy is channelled into reproduction than into other traits as *D. dianthus* is actively reproducing in shallow waters of this fjord in austral winter¹⁰⁷. The results of the present study indicate that deep corals generally have more energy available and are potentially also more fecund as proposed by Feehan et al.¹⁰⁷, but nothing is known about their reproductive cycle so far. Therefore, future studies on coral energetics should also include the reproductive cycle, in order to better understand the interplay of traits and potential fitness consequences for the corals as well as the whole population.”

Line 404-405: So is it possible that the corals transplanted from deep to shallow just still had more energy reserves left after a year so that they could afford keeping a higher tissue cover? Or do you have any other speculation why the shallow-origin corals could grow more tissue at the deep site, why can they not do that at the shallow site? Any speculation?

R: Yes, we think that novel corals in shallow waters still had more energy reserves left and were therefore able to maintain their larger tissue covered surface area after transplantation (line 449): “Deep corals transplanted to shallow were able to maintain their tissue covered surface area without tissue retraction (Supplementary Figure 7 and 8). Whether this is a real local adaptation or a delayed response, potentially caused by enhanced energy reserves, needs to be elucidated.”

However, we only have data about the tissue coverage of the novel shallow corals (Supplementary Figure 8) as the samples for the tissue analysis were thawed and we therefore have no biomass data for the transplanted corals from deep to shallow. It could be that these corals did not retract their tissue but reduced the tissue thickness, which would have resulted in a lower biomass. It would also have been interesting to see for how long they were able to maintain their tissue covered surface area. As we e.g. also mention the higher infestation rate of corals at shallow stations, we think that is a combination of different unfavourable factors at the shallow stations (e.g. higher environmental variability, likely linked to fluctuations in food availability, boring organisms, etc...) that prevents the shallow corals from investing a lot of energy into somatic growth and the build-up of energy reserves (line 442): “In addition, the low tissue cover is associated with a higher infestation with endolithic photoautotrophic

organisms^{103,104,105} (Supplementary Figure 7), which negatively affects their septal linear extension rates¹⁰⁵. This may be an additional stressor for shallow corals and contribute to a potentially reduced fitness¹⁰⁵ as the defence against infesting organisms and maintenance of the skeletal integrity requires an increased energy expenditure. However, this likely did not affect deep corals to the same extent, which were completely covered with tissue and therefore protected throughout the duration of the experiment.” We think that the situation at the deep station is more favourable for them and therefore, they are able to rapidly extend their tissue covered surface area after transplantation (discussion in line 365 and 472).

Conclusions

Line 437: environmental variability is increasing in the future, could you provide a reference?

R: Marine heat waves will become more frequent according to Frölicher et al. (2018) and Oliver et al. (2018), as is already stated in the discussion (line 346): “As extreme events are expected to increase in the future^{70,71}, higher environmental variability may expose CWCs to more stressful conditions.”

Frölicher, T. L., Fischer, E. M. & Gruber, N. Marine heatwaves under global warming. *Nature* 560, 360–364 (2018).

Oliver, E. C. J. *et al.* Longer and more frequent marine heatwaves over the past century. *Nat. Commun.* 9, 1324 (2018).

Reviewer #2 (Remarks to the Author):

This manuscript addresses a complex and important issue that will help further understand the risk to see cold water coral population disappear with global change. It is generally well-written and easy to understand. It is original and of high interest to the community of cold water coral researchers and broader biology community.

R: We thank the reviewer for the very positive feedback.

While this work is convincing and represents an impressive novel dataset, I have several questions that I did not find answers to from reading this manuscript:

1) Is it possible to claim that the fjord offers a natural gradient? Are conditions in the different sites really different from one another? I do agree with the difference between shallow and depth stations (lines 154-161), but it seems that it might sometimes be misleading to call the

head to mouth gradient a natural gradient without proofs that it actually is a gradient? (for eg. In Figure 2, we don't see this gradient).

R: We used the terms horizontal and vertical gradients based on previous data from Fillinger & Richter (2013) and the data shown in the Supplementary Table 1 of the present study. Here, temperature, salinity, oxygen concentration and pH show the clearest horizontal gradients in austral summer, while the transition and winter period show more variability. We added a brief note at the end of the introduction where the gradients are described (line 124): "The vertical gradient persists with the strong environmental differences described^{44,47,49}, whereas the horizontal gradient is strongest in the productive summer season, but influenced by mixing in autumn and winter (Supplementary Table 1, Supplementary Figure 3)." and also added a Supplementary Figure 3 with the seasonal data of the Supplementary Table 1 and the gradients.

Our Figure 2 only focuses on the depth gradient and we plotted the mean values of the shallow stations here on purpose as most of the parameters were measured only once per season and not continuously like temperature. However, we also included the single data points in this figure now and added the Supplementary Figure 4 with the data of all shallow stations.

New Supplementary Figure 3 (a-d: January, e-h: May, i-l: August).

Figure 3 from Fillinger & Richter (2013): station RB is at the head of the fjord and station LL at the mouth of the fjord.

Fillinger, L. & Richter, C. Vertical and horizontal distribution of *Desmophyllum dianthus* in Comau Fjord, Chile: a cold-water coral thriving at low pH. *PeerJ* 1:e194 (2013).

2) Why is there no inclusion of season in the LMM for respiration? (and the second model for growth). Clearly, the season is an important factor to take into account in the model. Or do the authors think that temperature is stable enough to have no seasonal effect? (Seasons are likely operating at least on food quality/quantity, if not directly on metabolic rates?).

R: As we had too many different factors for the small number of replicates, we splitted the statistical analyses into two models and only tested the differences between seasons and shallow stations in the first model (with all data of novel corals at shallow stations) and the transplantation effect and differences between water depths in the second model (including only native and novel corals of stations A, Es, Ed and F). We agree that seasonal differences are important and included the statistical output of model 1 for respiration in the tables in the supplementary material of the revised manuscript and also added this information now in the revised methods and results (line 880 and 200). In the second models for respiration and calcification, we only tested the difference between depths (E shallow and E deep) and between native and novel corals to find out if transplantation had an effect.

3) Similarly, why was there no “model 1” for shallow stations and respiration?

R: See answer to previous question. We added the results of model 1 for the respiration data of shallow stations in the revised manuscript.

4) It is difficult (with the material and methods at the end) to not get confused by the growth measurements, I wondered several times what was used in the manuscript. There are information mentioned in the material and method such as calcified growth or calcein staining that are not presented in the manuscript. I would advise the authors to reduce the material and methods by deleting all that information not presented in the results (e.g. lines 592-603, lines 523-524) and move it into supplementary.

R: We agree and unified terms (calcification rates instead of growth rates), clarified the information about the calcein staining (line 653): “Right before the first re-installation, the corals were stained with 50 mg l⁻¹ Calcein for 16-19 h in order to mark the beginning of the experiment in the skeleton for further skeletal analyses that are not part of this study.” We also moved the methods part about the “tissue corals” for biomass determination into the supplements as these results are only shown in the supplementary material.

5) A naïve question but can it be that the generally less fit individuals from B and C are affected by the nearby town activities? (e.g. sewage etc.).

R: There are no villages in the area. Settlements consist of a few (1-10) houses which are unlikely to have a profound effect on the fjord. A larger threat is salmon farming which introduces huge quantities of organic matter (Quinones et al., 2019). Salmon farms are located on both sides of the fjord, but coral stations were selected in some distance from major farms and stations were not chosen to be able to specifically address the impact of salmon farm activity on coral fitness. While stations C and D are located closest to salmon farms, stations B and F are furthest away from farms and therefore, corals at station B cannot be negatively affected by the salmon farm activity.

Quinones, R. A., Fuentes, M., Montes, R. M., Soto, D., & León-Muñoz, J. Environmental issues in Chilean salmon farming: a review. *Reviews in Aquaculture*, 11, 375-402 (2019).

Minor comments:

The introduction is well-written and lays the problematic clearly; it is good that terms such as acclimatization are clearly defined.

Results:

Figure 2B-E: I would have preferred to see each of the shallow stations (A-F) blue lines, as SD seems to be quite large (for pH for example – see Table S1). And maybe some of the shallow stations are closer to the deep station? It would also make the claim that it is a “gradient” better supported. From Table S1, we can see that station A is close to station Ed for pH. As it is a paper on variability, I would present the variability more clearly. It would also make clear that there is one measurement per station for each season, which should be clearer in the legend (since the material and methods are in the end).

R: We understand the point here and included a figure with the single data points for each station in the revised supplements (Supplementary Figure 4). As our main aim here is to show the differences between the deep and shallow stations, we only show the mean values (with standard deviations) in the main text to not distract from the clear difference in all parameters between the two water depths. However, we also included the individual data points in Figure 2 in the main text now to comply with the formatting guidelines of *Communications Biology*.

New Figure 2B-E:

Supplementary Figure 4:

Line 186: This 1st time use of “LMM” is a bit confusing, especially since it does not correspond to any acronym used in the material and methods either.

R: We included this acronym in material and methods now.

Line 189: The use of winter/summer can be confusing at first for people from the Northern Hemisphere, maybe it would be good that authors precise that the first time, e.g., “... in winter (august) compared to summer (January) and autumn (may)...”

R: Modified accordingly: “In shallow waters along the fjord, calcification rates were higher at the mouth of the fjord compared to the head and differed between seasons with lower calcification rates in austral winter (August) compared to austral summer (January) and autumn (May).”

Line 202: I personally believe that a figure similar to Figure S6 with temperature would be useful data for other scientists to be able to compare their data with (not just temperature variability).

R: We agree and now added two additional panels to the Supplementary Figure 9 with the mean annual and seasonal temperature on the X-axis.

Line 199: The p-value presented here is not the same as the one in Table S3. Is this a mistake? Or did I misread the results?

R: Thank you for pointing out the mistake. We corrected the p-value in the text.

Discussion:

Line 255: If I understood correctly, the “fittest” individuals are defined as the fastest growing? I would add this in e.g. ‘...found the fittest (i.e. fastest growing) *D. dianthus*...’ This is not the only definition of fittest and the reproductive output is also part of this equation, it seems that the fastest-growing individuals are also the individuals that have the highest metabolic rate, and if this was a consequence of a stress (not sure it is the case here), it could prevent individuals to invest in reproductive growth.

R: The reviewer is correct and we use the term “fittest” not in an evolutionary sense and from a population level point of view, but rather from an individual perspective and their measure of fitness. Nevertheless, calcification rate is also linked to reproductive output as the corals need to reach a certain size to start to reproduce and to escape spatial competition and increase the access to available food within a dense coral community. For more clarity, we

now added "...found the fittest (i.e. fastest growing) *D. dianthus*..." as suggested by the reviewer.

I would be actually interested in the ratio between respiration and somatic growth (how much respiration is needed to grow 1 mg: $\mu\text{mol O}_2$ per mg^{-1} – for each individual), as it also relates to the discussion on lines 320-324. I made a quick figure from the means provided and it looks like these ratios differ widely by station and by season (see figure attached). If this turns out to be interesting, I would recommend that it is added as a Figure 3C, it correlates with the result that B and C are in "trouble" (line 383) with high respiration rates for minimal growth rates.

R: We appreciate the reviewer's suggestion to compare respiration and calcification results. However, one caveat is that catabolic energy from respiration not only feeds into somatic growth (tissue growth) but into skeleton growth (calcification), reproduction, basal metabolism, etc. Solving linear equations with four (or more) unknowns requires four (or more) equations (which we don't have); the attempt to solve four unknowns with one equation is not possible. The wide variability in the respiration:calcification ratios between stations, seasons and depths is thus not surprising, as it indicates that three of the four unknowns (reproduction, somatic growth, basal metabolism) cannot be ignored. Another consideration is the different scale of integration between the parameters. Calcification integrates over months while the respiration incubations consider much shorter time scales (6 hours).

While not a needed addition to the discussion, the authors could be interested in a recent paper from Chilean environmental data (and variability) by Vargas et al. from 2022 in Nature Climate Change (12, 200-207).

R: Thank you for the suggestion. We included this study in the discussion (line 346).

Material and methods:

Line 483...: Was the transport temperature-controlled? Was the temperature the same as the sampling site? (also lines 522)

R: The temperature during transport was controlled passively by means of thermal insulation. Short distances (<1.5 h boat rides) and professional (Grade A) coolboxes kept temperatures within $<1^\circ\text{C}$ from *in situ* temperatures at the sampling sites.

In the flow-through aquarium system of the research station, corals were maintained in seawater that was pumped from 20-25 m water depth in front of the research station. Due to the high flow-through rate, it is expected that the temperature did not change during maintenance but corals were subject to natural temperature fluctuations of the fjord.

Line 559: It needs to be precised in the legend of Fig 3B that it is respiration rates at 12.5°C, it is not “true” respiration on site and in the site conditions. Why was respiration rates measured at 12°C and not at the temperature of the sampling site? Do we expect no difference due to 2-3°C differences?

R: We agree and now clearly mention in the methods section (line 697) and in the figure header (line 235) that the same temperature across stations and within seasons was used for the assessment of the respiration rates:

“Due to logistical reasons, the temperature of the water bath was controlled by a temperature-controlled room and small temperature differences (< 0.5 °C) between shallow stations were not taken into account as they are within the error of the thermostat. We also did not take into account the larger temperature differences between shallow and deep (up to 2 °C in summer and autumn) as we preferred to use uniform temperatures that deviate from *in situ* temperatures for reasons of standardisation and due to logistical reasons. Therefore, respiration measurements were conducted at standardised conditions, approximately representing water temperatures of shallow stations at the end of each season and seasonally increasing from 11.76 ± 1.63 °C in austral winter, 12.19 ± 0.63 °C in autumn and 14.16 ± 0.43 °C in summer (with more variable temperature conditions in winter as the temperature control system was not working well). This resulted in a seasonally standardised assessment of metabolic rates, which is not related to *in situ* temperature. Due to this standardisation and because we do not know the temperature performance curves of *D. dianthus*, the temperature change between *in situ* and incubation temperature can lead to either an under- or overestimation of the respiration rates of deep corals. Nevertheless, it allows a direct comparison of the metabolic potential of corals between stations.” (line 697)
“Respiration rates were measured at a standardised temperature of 12.75 ± 1.44 °C, representing water temperatures of shallow stations at the end of each season (January: 14.16 ± 0.43 °C, May: 12.19 ± 0.63 °C, August: 11.76 ± 1.63 °C) and not at *in situ* temperatures of each station.” (line 235)

How many days after collection were the respiration trials done? Had they acclimatized to laboratory conditions? Were the corals starved before starting the trials? All these are critical information for other scientists to be able to compare their results.

R: The respiration rate of the corals was measured immediately after re-collection. When they arrived in the laboratory, the corals were unscrewed from the holders, the screws and bare skeletal parts cleaned (took about 30 min) and the corals were then incubated. Therefore, the corals were not acclimatised to laboratory conditions and were not fed before the incubations. All this information is now added to the revised manuscript in the methods section (line 688): “Respiration rates were determined through closed-cell incubations immediately after re-collection of the corals without acclimatisation to laboratory conditions.”

Line 582: Was there a difference with different stations in skeletal aragonite density? If this was not tested (n=9), which stations were these specimens from?

R: We did not investigate spatial differences in skeletal density of *D. dianthus* in our study. The specimens for the density determination were from stations A and F, with overlapping density values. We included the information that corals were from stations A and F in the manuscript now (line 738): “Skeletal aragonite density for *D. dianthus* ($2.793 \pm 0.026 \text{ g cm}^{-3}$) was derived from nine experimental corals from stations A and F with overlapping density values after Davies¹⁰⁵.”

Line 662: I believe that the Environmental data paragraph should come right after the field site description, as it is the way the results are presented.

R: Modified accordingly.

Line 696-706: As said above, I have trouble understanding why the results of the first model are not presented for respiration in the manuscript? And why seasons are not included as a fixed factor for the second model?

R: See comment above in our answer to question 2 of reviewer 2 on the same aspect. We added the results of model 1 for respiration data of shallow stations now.

Line 711-716: I am not sure that I understand this part, is this single factor linear regressions (e.g. $\text{lm}(\text{growth} \sim \text{temperature})$ or $\text{lm}(\text{growth} \sim \text{temperature} + \text{salinity})$ and then it is compared by AIC)? I think this calls for a little more explanation on why choose these methods and what it means. Are the factors really linearly correlated? Maybe if these figures would be presented in the supplementary material, it would be easier to understand.

R: We used $\text{lm}(\text{growth} \sim \text{temperature variability} + \text{pH} + \text{aragonite saturation} + \text{salinity} + \text{oxygen} + \text{temperature})$. We performed model selection using Akaike information criterion (AIC), using the *AICcmodavg* package to step through all models from single environmental factors to multiple factors without inclusion of interactions. Overall, we expected to be in a range of all environmental variables where calcification is approximately linearly related and thus, used a linear regression approach. We provided more detail in the methods section (line 894): “We used AIC to go through all models from single environmental factors to multiple factors without inclusion of interactions. We used a linear regression approach here because we expected to be in a range of all environmental variables where calcification is approximately linearly related.”

Reviewer #3 (Remarks to the Author):

I reviewed with great pleasure the manuscript “Environmental stability and phenotypic plasticity benefit the cold-water coral *Desmophyllum dianthus* in an acidified fjord” of K.K. Beck and co-workers. ... Overall, the manuscript is of very good quality, language, figures and data presentation and suits very well to the journal’s scope. Therefore, I recommend publication with only very minor revisions:

R: We thank the reviewer for the very positive feedback and recommendation for publication.

1) I indicated in the annotated pdf, where I would include additional citation of references, which might have been overlooked, but I believe should be integrated to complete the picture and acknowledge these previous works.

R: We have integrated the recommended references and incorporated the proposed changes. However, we included White et al. (2012) and Guihen et al. (2018) in another part of the manuscript (lines 317 and 338) as they only found seasonal oxygen variability in CWC habitats, whereas we specifically talk about temperature variability in the sentence where the reference was suggested. We also stated more clearly now that apart from our study also Form & Riebesell (2012) found highest calcification rates at aragonite undersaturation in their long-term study (line 293): “Even though it was not statistically significant, a similar trend of highest calcification rates under aragonite undersaturation was also found in a long-term laboratory experiment with *L. pertusa*⁵¹, which is explained by the overall good physiological conditions of the corals due to regular feeding.”

2) I understand well that the authors focus mainly on the temperature variability in their study, as a strong correlation between temperature fluctuations and other environmental parameters has been observed (4,7,24,25,48). Nevertheless, I would like to see additional discussion about the similar negative correlation of pH or aragonite saturation state with growth (figure S6) especially having the “acidified fjord” in the title of the study.

R: We included some more discussion about the effect of pH and aragonite saturation on calcification rates (line 278): “CWC reefs in other areas (e.g. Gulf of Mexico, SW Australia) dominated by *L. pertusa* have also been found close to or below the aragonite saturation horizon^{6,7,9}. However, *D. dianthus* occurs at lower Ω_{arag} , pH, TA and DIC values than previously reported for CWC habitats in the NE Atlantic Ocean and Mediterranean Sea^{8,50}. In austral summer (January), the pH and aragonite saturation in shallow waters of the head of the fjord were as low as in deep waters at station Ed, presumably due to increased run-off and decomposition of terrestrial organic matter, showing that the observed differences in calcification rates are not related to differences in aragonite saturation (Supplementary Figure 9a-d). As we found fittest specimens at highest DIC values in Comau Fjord, this does

not support the hypothesis that the occurrence of healthy CWC reefs is prevented by high DIC⁸. While breakage and dissolution of bare coral skeletons occurs under aragonite undersaturation¹⁵, there is growing evidence that carbonate chemistry is not as important for live CWCs as has long been thought. Several laboratory and field studies have confirmed the capability of CWCs to calcify and survive at aragonite undersaturated conditions^{15,24}. Even though it was not statistically significant, a similar trend of highest calcification rates under aragonite undersaturation was also found in a long-term laboratory experiment with *L. pertusa*⁵¹, which is explained by the overall good physiological conditions of the corals due to regular feeding.“

This comparable correlation exists for the months of May and August - but not January. Why is this the case? Is that a result of large T-variability in January compared to May and August? Could this be related to better (winter) mixing in January?

R: January represents austral summer here and thus, this cannot relate to winter mixing. In January, the pH is as low at the head of the fjord as at the deep station (presumably due to increased melting and run-off of terrestrial organic matter decomposing at depth in summer), which is why the stations with lowest and highest calcification rates of corals coincide in January (Supplementary Figure 9b and d). Therefore, this does not permit a correlation of calcification rates with pH and supports the notion that pH is not a main driver here (we also included this aspect in the discussion now, line 281). Even though temperature variability is high in December, it is much lower during the rest of the austral summer season (September to November), which is why the seasonal temperature variability is also lower at shallow stations in summer compared to autumn and partly also winter (stations at the head of the fjord; Fig. 4b).

Both parameters (or the whole parameters of seawater carbonate chemistry) are indispensable for CWCs during the calcification process as e.g., *Desmophyllum pertusum* mainly relies on an external DIC source for the calcification process (Mueller et al., 2014). Further, it had been suggested that CWCs can increase their internal DIC concentration (McCulloch et al., 2012), but strongly elevated seawater DIC can possibly not be regulated by metabolic/calcification processes (Flögel et al., 2014) - a closer look to the CO₂ system is therefore essential and should be included while considering fitness, growth or respiration of CWCs. As the authors present this comprehensive data set including all seawater carbonate chemistry parameters, I would recommend not to omit these aspects important for CWC's adaptability and include them into the discussion (as additional paragraph) and conclusion.

R: We agree with the reviewer that carbonate chemistry is important. Certainly, it would be interesting to better understand how different carbonate chemistry parameters influence calcification and could be more beneficial than others (additionally if linked to different levels of resource availability). However, Flögel et al. (2014) discuss that CWCs are limited at 2170

μmol but in our study, corals under elevated DIC (so not exceeding 2170 μmol) are rather fitter and faster growing than corals that occur at lower levels. Thus, this is not in full support of this statement and supports the assumption of the present study that the carbonate chemistry is not the main driver for CWC occurrence and fitness in Comau Fjord.

In addition, we revealed that other factors are more important for our study site. We clearly show that temperature variability correlates with other parameters and we could recently also show that pH fluctuates with temperature and salinity changes in the fjord. We therefore think that all parameters are subject to strong variability in shallow waters, which affects the physiological response of the corals. Thus, in our conclusion we emphasized the role of environmental variability that includes a range of environmental parameters. As we also state in the manuscript, we think that the high food availability in this region is beneficial for the corals and might enable them to thrive at aragonite undersaturation. Therefore, carbonate chemistry may not be such a prominent driver here as it would be in regions with limiting food supply.

REVIEWERS' COMMENTS:

Reviewer #2 (Remarks to the Author):

I want to thank the authors for thorough answers and the many corrections/clarifications made in the manuscript that help make it much clearer now. I want to thank the authors in particular for making their "raw" data available through the many supplementary figures and tables supplied.

I am especially happy with the new modified figures showing individual variability more clearly and Figure 2 accompanying supplementary (Fig. S4). The natural gradients are now clearly defined without needing to revert to other articles. The replacement of "growth" by "calcification" now makes what was measured clearer. I understand much better the choices of models and the model selection process.

I am still a bit doubtful about the respiration measured just a few hours after leaving their natural habitat (at a different temperature), but as this is now clearly stated (in figure legends and in the extra discussion), the readers will be advised to regard these results with caution if they would like to compare it with their own or use the numbers for e.g. modeling work. This statement will also help better understand why measured respiration rates can be so different when comparing different (or sometimes even the same) CWC species.

Regarding Figure S9 and calcification vs. temperature, for future work I think it is interesting that seasonal mean temperature is positively correlated in May (higher temperature = higher calcification) but negatively in winter (higher calcification at the lowest temperature).

Regarding my suggestion to compare respiration and somatic growth, I believe I did not express my thoughts well-enough when writing my comment. I was specifically thinking about "the rest" (as you said, calcification, reproduction, metabolism) so the equation will be respiration = somatic growth + "the rest" (including losses). What I would have found interesting there is that if for example you would observe in a group a higher respiration with similar somatic growth than in another group, you could suggest that, *maybe*, a part of this respiration for this group is diverted for other processes such as basic metabolism, or more energy spent in catching food.

This is what you see for example in some organisms that increase their respiration rate in acidified seawater: they do not grow faster but they respire more (when given non limited amount of food), and this indicates that extra energy is needed to maintain the same processes/growth rate (maybe through increased ionic exchanges with the seawater, see for example Stumpp et al. 2012, PNAS: <https://doi.org/10.1073/pnas.1209174109>). But this was just a personal interest and it might not fit well in this manuscript (and I agree with the problem of scale integration for this experiment).

Again, this is impressive work and I can only commend the authors for their care in making the replies very informative.

I believe their manuscript is ready for publications without the need of further changes.

Dr. Narimane Dorey